# Glacier Mass Balance and Its Response to Heatwaves for Kangxiwa Glacier in the Eastern Pamir: Insights from Time-Lapse Photography and In-situ Measurements

Ying Xie<sup>1,2</sup>, Baiqing Xu<sup>1,2</sup>, Meilin Zhu<sup>3</sup>, Yu Fan<sup>1,4</sup>, Pengling Wang<sup>5</sup>, Song Yang<sup>6</sup>, Wenqing Zhao<sup>1,4</sup>, Wei Yang<sup>1,2\*</sup>

<sup>1</sup>State Key Laboratory of Tibetan Plateau Earth System, Environment and Resources (TPESER), Institute of Tibetan Plateau Research, Chinese Academy of Sciences, Beijing 100101, China

<sup>2</sup> Muztagh Ata Station for Westerly Environment Observation and Research, Chinese Academy of Sciences, China

<sup>3</sup> College of Earth and Environmental Sciences, Lanzhou University, Lanzhou 730000, China

<sup>4</sup>College of Resources and Environment, University of Chinese Academy of Sciences, Beijing 100049, China

<sup>5</sup> National Climate Centre, China Meteorological Administration, Beijing 100081, China

<sup>6</sup> School of Geographical Sciences, China West Normal University, Nanchong 637009, China

Correspondence to: Wei Yang (yangww@itpcas.ac.cn)

Abstract. Contrary to the widespread glacier mass loss in High Mountain Asia under global warming, glaciers in the Pamir-15 Karakoram region have exhibited anomalous less mass changes and even slight mass gains in recent decades. While geodetic studies have quantified decadal-scale mass loss, the process of glacier mass balance and its response to regional climate change remain poorly understood due to the scarcity of high-resolution observations. This study analyzes the characteristics of daily glacier mass balance and their responses to the heatwaves based on time-lapse photography, ablation stake/snow pit measurements and meteorological data collected at the Kangxiwa Glacier in the eastern Pamir. Our results showed that the Kangxiwa Glacier experienced weak mass loss in 2019/2000 and 2020/2021 balance years but significant mass deficits in 2021/2022 and 2022/2023. Observations evidence that the Kangxiwa Glacier is a spring-accumulation summer-ablation type, with spring (April-June) accumulation of +200-500 mm w.e. and summer (July-September) mass loss of 300-900 mm w.e. The unprecedented heatwave in July-August 2022 caused an abnormal mass loss of over -852 mm w.e. within 40 days, advancing the Glacier Mass Loss Day by one month and pushing the equilibrium line altitude above the glacier summits. The 2022 heatwaves, characterized by wakened westerly circulation, likely influenced not only the East Pamir region but also the western Kunlun Mountains, leading to increased incoming radiation and reduced precipitation. Our studies revealed that the high-elevation glaciers in eastern Pamir are sensitive to the heatwaves, suggesting that the termination of the socalled Karakoram anomaly may reflect recent climatic warming in this high-elevation region.

# 1. Introduction


Under global climate warming, glaciers on Tibetan Plateau and its surrounding regions have suffered from significant mass loss over recent decades (Brun et al., 2017; Bhattacharya et al., 2021; Hewitt., 2011; Hugonnet et al., 2021; Shean et


al., 2020). However, the notable exception, called the "Pamir–Karakorum" anomaly, occurred in the western Kunlun, Karakoram and the eastern Pamir ranges, where glaciers have remained in balance or experienced slight mass gains since at least the 1970s (Hewiit et al., 2011; Berthier and Brun, 2019; Brun et al., 2017; Kääb et al., 2015). Recent studies suggested that this anomaly maybe be transitioning to a generalized thinning, indicating the end of the 'Pamir–Karakorum' anomaly (Hugonnet et al., 2021).

Glaciers in Central Asia are critical component of the hydrological cycle, providing substantial runoff during the dry summer months for agriculture or hydropower (Huss and Hock, 2018). In-depth analysis of glacier mass changes and their climate response is therefore essential for water resource management and regional sustainability. Geodetic studies have successfully determined decadal glacier mass changes (Shean et al., 2020; Hugonnet et al., 2021). However, the heterogeneous climatic and topographical conditions across Tibetan Plateau and surroundings result in high spatial variability in glacier mass balance (Brun et al., 2019; Barandun et al., 2021; Barandun and Pohl, 2023; Zhu et al., 2023). In addition, long-term glaciological measurements are scarce in the western Kunlun, the Pamir and Karakoram ranges (Barandun and Pohl, 2023; Yao et al., 2022; Zemp et al, 2023). Direct glaciological measurements using ablation stakes and snow pits enable the derivation of seasonal and annual mass balance, supporting model calibration and validation (Zhu et al., 2018; Ren et al., 2018). The limited availability of in-situ observations and detailed analysis of physical ablation and accumulation process has hindered a comprehensive understanding of the factors driving these glacier dynamics.

Recent extreme events such as heatwaves and droughts have caused abnormal high-elevation melting (Chen et al., 2023; Gui et al., 2024; Hassan et al., 2024; Little et al., 2019), threatening water security and triggering glacier-related disasters (Kääb et al., 2018; Shugar et al., 2021; Zhao et al., 2022). For examples, a 25-day heatwave in Switzerland in 2022 caused melt equivalent to 35% of the total summer ablation (Cremona et al., 2023). Similarly, the 2022 heatwave induced unexpected melting on the central Tibetan Plateau (Zhu et al., 2023) and severe mass loss at Urumqi Glacier No. 1 in the eastern Tien Shan (Xu et al., 2024). The increasing intensity, frequency and duration of regional extreme events around the world pose a significant threat to mountain glaciers (Oliver et al., 2018; Perkins-Kirkpatrick and others, 2020). Traditional glaciological method and geodetic survey can provide mass balance data at multi-year, annual, and seasonal scales. However, both approaches face challenges in capturing the high-resolution evolution of surface mass balance dynamics, limiting our understanding of glacier response to short-term extreme events.

Recent advancements in high-temporal-resolution monitoring techniques, such as steel wire anchored below the ice surface (A2PS contributors 2021), automated cameras monitoring colour-coded ablation stakes (Landmann et al., 2021; Cremona et al., 2023), terrestrial laser scanning techniques (Voordendag et al., 2023), have provided new insights into short-term mass balance variations, including their response to the extreme melt events (Cremona et al., 2023). New indices like the Glacier Loss Day (GLD), which was defined as the day when net mass balance becomes negative and winter snow is exhausted (Voordendag et al., 2023), have been developed to characterized the glacier-climate imbalance. In this study, we applied time-lapse photography at the Kangxiwa Glacier from 2019–2023, combined with in-situ glaciological and

meteorological measurements, to analyze the characteristics of daily accumulation-ablation processes, interannual mass balance differences, and to investigate the impact of extreme heatwaves on high-elevation glaciers in the eastern Pamirs.

# 2. Study regions

The study area is located near Muztagh Ata (7546 m a.s.l.) and Kongur Shan (7719 m a.s.l.) in the eastern Pamir, Central Asia (Fig. 1). This region hosts a total of 434 glaciers covering 1018 km², with a continental glacier regime characterized by cold and arid conditions (Shi & Liu, 2020). The climate of eastern Pamir is dominantly controlled by the westerly jet stream (Yao et al., 2012). Data from the Taxkorgan Meteorological Station (3091 m a.s.l., ~50 km south of Muztagh Ata) show a mean summer temperature (June–August) of 15.1°C and annual precipitation of ~70 mm (1965–2023), concentrated between April and September.

Kangxiwa Glacier (38.28°N, 75.28°E) is a debris-free valley glacier on the western slope of Kongur Shan, extending from 5350 m to 4960 m a.s.l. with an area of 1.86 km² and length of ~3 km. This glacier was selected as the benchmark glacier for long-term measurements (Yao et al., 2022). Geodetic estimates indicate a near-balanced mass state of surrounding glaciers in the eastern Pamir with +0.13±0.10 m water equivalent (w.e.) during 2000–2019 (Hugonnet et al., 2021), with moderate mass loss of -0.07±0.20 m w.e. during 2019–2022 (Falaschi et al., 2023) and a low area shrinkage rate of 0.7±0.5% decade<sup>-1</sup> (Li et al., 2022).

**Figure 1.** Study region and the distribution of in-situ measurements. (a) Location of the studied area in the eastern Pamir, central Asia; (b) The location of Kangxiwa Glacier (red triangle), Automatic Weather Station (AWS) at the elevation of 4900 m and Taxkorgan meteorologic station (red stars); the World hillshade layer was used as the background (http://www.arcgis.com) and the outline of the glacier was obtained from the Randolph glacier inventory (RGI 7.0; http://www.glims.org/RGI); (c) Photograph of the time-lapse camera and colour-coded stake on the Kangxiwa Glacier; (d) Topographic map of the Kangxiwa Glacier showing the locations of three time-lapse camera monitoring systems and the ablation stakes/snow pits for in-situ mass balance observations.

#### 3. Data and Methods



# 3.1 Surface mass balance measurements by stakes and snow pits

Ten ablation stakes were installed along the central flow line of Kangxiwa Glacier to measure point-scale mass balance following standard glaciological methods (Yu et al., 2013). Field measurements of stake height, snow stratigraphy, and





density were conducted at the start (early June) and end (late September) of each ablation season (Fig 1d). Ice density was assumed to be 900 kg/m<sup>3</sup>. The winter, summer and annual point mass balance were calculated during the period from 2019 to 2023. These annual and seasonal mass balances during 2019–2023 serves as ground truth for validating time-lapse camera data in this study.

# 5 3.2 Surface mass balance monitoring by time-lapse camera

# 3.2.1 Time-lapse camera monitoring systems

Three time-lapse camera systems were deployed at the glacier terminus (5005 m a.s.l.), mid-glacier (5137 m a.s.l.), and accumulation zone (5300 m a.s.l.) (Fig 1d). The monitoring system adhered to methodologies established in prior research (Landmann et al., 20221; Cremona et al., 2023), integrating color-coded aluminum stakes and time-lapse cameras to quantify the change in stakes' height. The aluminum stakes were marked at 5 cm intervals with alternating red and yellow bands, installed vertically on the glacier surface and photographed hourly by fixed-position Forsafe H801 time-lapse cameras. Powered by 8 AA batteries and a solar panel, the cameras ensured continuous field operation. Featuring waterproof specifications, the camera captured images in JPEG format and stored them on a microSD card. In this study, cameras at 5137 m and 5300 m began operation on October 1, 2019 and the camera at 5005 m started on June 20, 2020. Seasonal maintenance included the replacements of stakes, microSD cards and batteries.

# 3.2.2 Changes of stake hight

A semi-automatic procedure was developed to measure the hight of the coloured stake, focusing on addressing the challenge of illumination variation in natural environments. The Hue-Saturation-Value (HSV) colour space was used for image processing because it separates luminance from colour information, unlike the standard RGB colour space where intensity and colour are intermingled. In the HSV method, H represents the colour type, such as red, yellow or blue; S refers to the purity of a colour, describing how much a pure colour is diluted with white light; and V represents the colour's brightness, ranging from black to white. These characteristics make HSV particularly suitable for analysing and segmenting images captured under diverse lighting conditions (Ganesan et al., 2014; Hamuda et al., 2024; Yu et al., 2021).

Prior to image processing, photographs taken between 12:00 and 19:00 were selected to ensure optimal lighting, and blurry images taken during heavy snowfall were manually excluded. A fixed frame was then defined to encompass the stake, leveraging its stable position across consecutive shots (see Fig. 2a). Stake contour detection relied on exploiting the difference in saturation values between the stake and the glacier surface (Fig. 2b). This involved applying a Gaussian filter to suppress high-frequency noise, followed by a morphological top-hat filter on the S channel of the HSV colour space to enhance differentiation between the stake and the background. Otsu's method (1979) was used for automatic thresholding during S-channel binarization, with the largest connected region selected as the stake in order to eliminate minor segmentation artefacts.


The pixel length of the stake  $(L_p)$ , which is defined as the height of the minimum bounding rectangle (see Fig. 2c), was converted to a real-world length in centimeters (L) using a linear relationship. This relationship was established through a comparative analysis of  $L_p$  and L in reference images. Reference images were manually selected to cover the full measurement range of  $L_p$ . For each reference image, L was manually calculated by incorporating the total number of visible colored bands (N), the pixel length of the bottom section  $(L_{pb})$ , and the pixel lengths of the two adjacent upper bands  $(L_{pa})$ , which were manually counted and measured, respectively. Figure S1 shows the significantly strong linear relationships between L and Lp of the reference images for each monitoring period at each monitoring site.

$$L=(N-1)\times 5 + L_{pb}\times 10/L_{pa} \tag{1}$$

Figure 2. Illustration of image processing. (a) The original image with a frame; (b) the grayscale representation of the S channel; (c) the contour of the scale stake with the minimum bounding rectangle in red.



**Figure 3.** The performance of stake changes derived automatically from time-lapse cameras (black line), compared with in-situ observations (red square) and manual calculations (blue triangle) for a stake at the elevation of 5005 m asl. Noted that the camera monitoring systems were maintained in June and September in each year during the period from 2020 to 2023.

The performance of stake changes derived automatically from time-lapse cameras was validated by comparing with insitu stake observations and manually calculated results (Fig. 3). During the image selection for manually reading, the reference images were excluded to ensure the independence of the validation dataset. The semi-automated procedure demonstrated strong agreement with both methods, with a mean difference of -0.30 cm across 56 manual validation samples, and a range of -2.7 to 0.18 cm (mean -0.73 cm) for in-situ readings. These comparisons highlight the robustness of the monitoring stake hight changes by time-lapse cameras, which has broader applications in studying surface mass balance dynamics at multiple elevations (Fig. 4).


**Figure 4.** Changes of stake hight at three elevations of the Kangxiwa Glacier during the hydrological years from 2019/2020 to 2022/2023. The stake hight was set to 0 cm at the beginning of each hydrologic year.

#### 150 3.2.3 Glacier mass balance estimation from change in stake hight

The daily changes in the hight of the color-coded stakes were converted into changes in mass balance by multiplying by the corresponding density for the surface conditions (Fig. S2). For bare ice, a density of 900 kg/m³ was used. For snow surfaces, a mean snow density of 450 kg/m³ was applied (the average density of snow pits measured at the three monitoring sites in June over the period 2020–2023). To account for uncertainty arising from variations in snow density, minimum and maximum values of 300 kg/m³ and 600 kg/m³, respectively, were incorporated into the analysis. Seasonal mass balance observations from traditional glaciological methods were used to validate the daily estimates from time-lapse cameras (Fig. 5). The mean differences of the 3 points are -71.7, -13.0 and -16.2 mm w.e., with standard deviations of 174.0, 89.1 and 134.0 mm w.e. respectively, confirming the reliability of the camera-based approach. *Overall, glacial mass balance changes calculated using the camera-based method are lower than those derived from field measurements. This is particularly notable during the 2022/2023 ablation season at the 5005 point, where the difference reached -409.02 mm w.e.—the largest* 


discrepancy observed. This is likely because ice and firn were mixed on the glacier surface during this period; however, we identified the surface material as firn based on the images. This misidentification led to an underestimation of surface material density, which in turn resulted in an underestimation of surface mass loss.

Glacier-wide daily mass balance was then derived via a weighted average of point values, with weights assigned by elevation-specific area proportions. Digital elevation model (DEM) data (http://www.gscloud.cn/) with a resolution of 30 m × 30 m were used to partition different elevation ranges. The three measurement points represented elevation ranges of 4960-5080 m (weight: 0.33), 5080-5200 m (weight: 0.38), and 5200-5390 m (weight: 0.29). Gaps in camera data were filled using daily mean mass changes for the corresponding periods. For example, the net change in mass balance during July 26 to September 25, 2022 at the 5300 site was -87.61 mm w.e. (450 kg/m<sup>3</sup> multiplied by the net change in stake height), the 170 daily average mass balance was-1.41 mm w.e. Notably, the 5005 m site was inactive from 1 October 2019 to 20 June 2020. During this period, daily average mass change of the in-situ measured values were used for glacier-wide calculations.

**Figure 5.** Comparison of the accumulated mass balance estimated using time-lapse cameras (black lines) and the glaciological methods (blue squares) at the three locations (5005 m, 5137 m and 5300 m) on the Kangxiwa Glacier.

### 3.3 Meteorological stations and the reanalysis data


An automatic weather station (38.28°N, 75.04°E, 4900 m a.s.l.; Fig. 1) was deployed on the western slope of Muztagh Ata in 2011 at a similar elevation to the Kangxiwa Glacier terminus. It continuously recorded half-hourly measurements of wind speed, wind direction, air temperature, relative humidity, atmospheric pressure, incoming/outgoing shortwave and longwave radiation, and precipitation (Zhu et al., 2018). This dataset was used to characterise the near-surface

meteorological features of the study area over the past four years and identify extreme weather events. This provides a basis for explaining short-term abnormal changes in glacier mass balance. Data from the Taxkorgan Meteorological Station (3091 m a.s.l.), located approximately 50 km south of Muztagh Ata, were used to analyze long-term climate change trends in the East Pamir. This situates the four-year study period within a decadal climate context.

The study also employed the fifth generation of reanalysis data from the European Centre for Medium-Range Weather Forecasts (ERA5), which has been widely used in climate research (Hoffmann et al., 2019) and glacier change analyses (Zhu et al., 2024). By analyzing geopotential height, wind fields, and surface air temperature, we investigated how large-scale atmospheric circulation influences extreme weather events and abnormal glacier ablation.

# 4. Results





#### 4.1 Evolution of glacier surface mass balance during the 2019/2020-2022/2023 hydrological years

Figure 6 illustrates the daily cumulative changes in the surface mass balance of the Kangxiwa Glacier during the 2019/2020–2022/2023 hydrological years, which can be broadly categorised into three phases: a balance period from October to mid-April, a snow accumulation period from mid-April to June/July, and an intense ablation period from June/July to September (Figs. 6 and 7). During the balance period, low temperatures and low precipitation resulted in limited changes in stake hight, with the cumulative mass balance fluctuating between -80 and +185 mm w.e. The slight surface mass loss during this period was primarily caused by sublimation or mechanical snow erosion by wind, which was evidenced by the abrupt transition from snow-covered surfaces to exposed bare ice (Fig. S2). During the accumulation period (mid-April to June/July), the glacier experienced substantial snow accumulation. At an elevation of 5300 m asl, maximum snow depth reached 1.0-1.5 m (Fig. 4), while the average snow accumulation across the entire glacier ranged from 270 to 410 mm w.e. Notably, monthly accumulation peaks generally occurred in May (Fig. 7).

Figure 6. The contrasting evolution of the cumulative glacier-wide mass balance of the Kangxiwa Glacier with daily precipitation (a-d) and the accumulated positive degree recorded at AWS4900 (e) during the 2019/2020–2022/2023 hydrological years. The 2021/2022 hydrological year, which own significant mass loss, is highlighted in red.

The phase of glacier mass loss is predominantly confined to the period from July to September, with significant interannual variability (Figs. 6, 7). This variability is evident in several aspects, beginning with the initiation of surface mass loss, which exhibits substantial interannual differences. The earliest commencement was observed on around 2 June 2023, while the latest occurred around 28 July 2020. Concurrently, the Glacier Loss Day (GLD)-defined as the date when the net mass balance transitions to negative and all winter snow accumulation is depleted (Voordendag et al., 2023)-occurred on



around 11 July 2022, approximately one month earlier than in other years (e.g., ~20 August 2020; ~3 September 2021; ~26 August 2023). In addition to the timing of initiation, the duration of glacier mass loss varies significantly from year to year. Specifically, it was approximately 30 days in 2020 (from ~28 July to 30 August), 80 days in 2021 (11 July to 30 September), 58 days in 2022 (26 June to 22 August), and 120 days in 2023 (2 June to 30 September). Another dimension of variability lies in the magnitude of annual surface mass loss, which ranges from -400 to -957 mm w.e. The most significant mass loss occurred in July–August 2022 (Fig. 7), with the mass loss in July 2022 being four times greater than the three-year mean for the same period. AWS records indicate that interannual fluctuations in thermal conditions were the primary driver of significant differences in both the duration and intensity of surface mass loss during the 2019/2020–2022/2023 hydrological years (Fig. 6e). The exceptional mass loss in 2022, for instance, coincided with anomalous summer warming, as demonstrated by the fact that accumulated positive degree days (CPDDs) at AWS4900 reached 341.7 °C·day, which is 1.4-2.3 times higher than in the other three years.

Figure 7. Comparison of monthly glacier mass balance of Kangixwa Glacier between four hydrological years of 2019/2020-2022/2023.

The annual mass balance of the Kangxiwa Glacier exhibited significant variability throughout the study period. A slight positive balance was recorded in the 2019/2020 hydrological year (+91 mm w.e.), followed by a weak negative balance in 2020/2021 year (-120 mm w.e.). This contrasts sharply with the substantial mass loss observed in the following two years: -707 mm w.e. in 2021/2022 and -335 mm w.e.in 2022/2023. The mass changes during 2019/2020-2020/2021 are consistent with previous observations of limited mass loss in the Pamir region (Yao et al., 2012; Falaschi et al., 2023; Bhattacharya et al., 2021; Hugonnet et al., 2021). Meanwhile, linear altitudinal interpolation of camera-derived point mass balance data across three elevations enabled estimation of the mean equilibrium line altitude (ELA) at ~5250 m a.s.l. over the study

period. This ELA, however, displayed significant interannual fluctuations: during 2021/2022, the highest ELA surpassed the glacier summit, whereas the lowest ELA, at ~5091 m a.s.l., occurred in 2019/2020.

#### 4.2 Glacier response to the unprecedented heatwaves in the summer of 2022

Both meteorological stations and ERA reanalysis data revealed unprecedented summer warming in the East Pamir, particularly in July 2022 (Fig. S3). The daily maximum temperature at Taxkorgan station in summer 2022 was significantly higher than the mean between 1957 and 2023, with 61% of days in July exceeding the 90th percentile and meeting the criteria for an extreme heat event defined by Lu et al. (2024). The ERA data also confirmed that the corresponding grid of the Kangxiwa Glacier in July 2022 was the highest recorded between 1980 and 2023. This provides a unique opportunity to analyze how glaciers respond to extreme heatwaves in the eastern Pamir. Based on AWS4900 records (Fig.1), three extreme heat events were identified between 26 June and 11 July, 18–30 July, and 5–17 August 2022 (Fig. 8).

Figure 8. Comparison of (a) daily air temperature, (b) daily incoming shortwave radiation, (c) daily precipitation and (d) daily glacier-wide mass balance at the Kangxiwa Glacier during the ablation season (June–September), showing the difference between 2022 and the other three years. The grey line shows the average for 2020, 2021 and 2023 with standardized variation (grey shaded area), while the red line shows the records for 2022. Light red rectangles highlight three heat events.

During the first two heatwaves, daily temperatures were 6.2°C and 4.4°C higher than the three-year average, with a maximum anomaly of +10°C on 5 July (Fig. 8a). Concurrently, daily incoming solar radiation exceeded the average by 90.6 W/m² (+31%) and 48.7 W/m² (+17%), while total precipitation decreased by 40.4 mm (-100%) and 31.3 mm (-76%) (Table 1). These conditions, characterized by increased temperatures, reduced snow replenishment, and intensified solar radiation, drove anomalous mass loss (Figs. 6, 7). The first extreme event induced a mean mass loss rate of -17.9 mm w.e./day,

whereas the three-year average for the same period ranged from -1.1 to +1.3 mm w.e./day, with a mean value of only -0.2mm w.e./day. The second heatwave exacerbated mass loss to -20.0 mm w.e./day, which is ~4 times the three-year mean (-1.0 to -8.0 mm w.e./day). This unprecedented ablation accelerated the depletion of spring-accumulated snow (Fig. 9) and thus contributed to the advance of the Glacier Loss Day by one month (Fig. 6).

Figure 9. Contrasting glacier surface conditions at the elevation of 5005 m between July-August 2020 and 2022.

The weather conditions during the August heatwave and the response of glacier were obviously different from those during the two July heatwaves. The magnitude of warming in the August heatwave was only 2.5 °C higher than the corresponding temperature average of the other three years. The reduced shortwave radiation of 54.9 W/m² (-17.1%) and the increased longwave radiation of 26.1 W/m² (+10.2%) indicate the increased cloud cover over the eastern Pamir during this period (Liu et al., 2021). There was no significant change in precipitation (Table 1). The August heatwave was therefore characterized by the relative warming and cloudy conditions. In contrast, the magnitude of mass loss during the August heatwave was even greater than those during the two July heatwaves (Fig. 8b). Mass loss intensity increased to be 23.3 mm w.e./day during the August heatwave, which is 15.6 mm w.e./day higher than the mean value for the corresponding periods of the other three years. Givern the limited magnitude of warming and the reduced atmospheric radiations, this discrepancy could partly be attributed to the lowest surface albedo of exposed dirty ice (Fig. 9), enhancing solar radiation absorption for melting (Mölg et al., 2014).

**Table 1** Daily anomalies of meteorological variables and mass balance during three heatwave periods in 2022 relative to the corresponding 2020–2021–2023 average.

| Phase          | Temperature | Shortwave                     | Precipitation | Longwave radiation | Mass balance |
|----------------|-------------|-------------------------------|---------------|--------------------|--------------|
|                | (°C)        | Radiation (W/m <sup>2</sup> ) | (mm)          | $(W/m^2)$          | (mm w.e/day) |
| I (6.26-7.11)  | 6.3         | 90.6                          | -2.5          | -11.2              | -17.7        |
| II (7.18-7.30) | 4.4         | 48.7                          | -2.4          | -10.9              | -15.0        |
| III (8.5-8.17) | 2.5         | -54.9                         | -0.4          | 26.1               | -15.6        |


The relationships between the monthly mass balance and the corresponding cumulative positive temperatures and precipitation further confirm the nonlinear responses of the glacier mass balance and the importance of surface conditions (Fig. 10). In months with similar positive temperatures (e.g., August 2020, July 2021, August 2022 and August 2023), the magnitude of glacier mass loss varied significantly (-130 to -380 mm w.e.), whereas precipitation varied only within 100 mm. This indicates that differences in precipitation alone cannot explain these discrepancies. Similar to the August 2022 extreme loss, glacier surface conditions (e.g., albedo) at different periods may play a critical role in modulating the energy balance. The timing and intensity of heatwaves have a differential impact on glacier mass loss, which is mediated by snow/ice albedo feedback mechanisms.

**Figure 10.** Relationships between monthly glacier mass balance, cumulative positive degree days (CPDDs), and monthly precipitation during the 2020–2023 ablation seasons. The anomalous negative mass loss in July 2022 is highlighted by a red circle.

#### 5. Discussions



#### 5.1 Mass balance characteristics in the East Pamir

Unlike the traditional seasonal or annual glacier mass balance observations (Zhu et al., 2018; Xu et al., 2024), this study presents the four-year records of daily-averaged glacier mass balance for a benchmark glacier in the East Pamir. Based on this dataset, it is evident that the accumulation in April–May is the primary determinant of annual accumulation magnitude. Barandum et al. (2018) found that glaciers in the West Pamir region that initiate snow accumulation at the start of the hydrologic year receive over 1 m w.e. of snow accumulation during the winter season. These different accumulation patterns are linked to divergent precipitation seasonality between the East and West Pamirs, driving the formation of distinct mass








accumulation types (Maussion et al., 2012). Published geodetic estimates reveal striking west-east disparities in glacier mass balance since 2000, with a mean mass loss of -0.26 m w.e. a<sup>-1</sup> in the Western Pamir but a near-balanced state (-0.02 m w.e. a<sup>-1</sup>) in the Eastern Pamir (Bolch et al., 2019). These contrasting mass loss patterns are likely modulated by divergent precipitation regimes and topographic conditions, which influence mass change patterns and climatic sensitivities across the Pamir (Brun et al., 2019; Wang et al., 2019).

Four-year observations revealed that the interannual accumulation differences of the Kangxiwa Glacier were limited to be ~260 mm w.e., with the maximum accumulation of 537 mm w.e. in the 2019/2020 hydrological year and the minimum of 278 mm w.e. in the 2021/2022 hydrological year (Fig. 6). By contrast, annual mass balance fluctuations occurred between a slight positive balance of +79 mm w.e. in the 2019/2020 and a significant negative balance of -702 mm w.e. in 2021/2022, resulting in a divergence of 781 mm. A comparative analysis between the 2019/2020 and 2021/2022 reveals that accumulation variability accounts for ~33% of the total mass balance difference, while differences in summer ablation contribute ~67%.. These results emphasize that interannual variability in the glacier mass balance in the eastern Pamir was predominantly driven by the variability in mass loss during the ablation season.

As illustrated in Figures 6 and 7, the mass loss in 2019/2020 was concentrated in August (-400 mm), following a period of high accumulation (+537 mm). In contrast, the 2022 mass balance process featured low spring accumulation and extremely strong summer ablation driven by heatwaves in July and August. While the ablation periods in 2021 and 2023 were significantly longer than in other years, the average summer ablation intensity was moderate. Unlike the stable mass accumulation observed in June across the other three years, the Kangxiwa Glacier experienced early ablation (-100 mm w.e.) in June 2023, followed by the highest ablation in September over the four-year period. Cumulative mass loss from June onward reached -880 mm w.e. in 2023, which was slightly less than the maximum loss in 2022. These divergent patterns of mass balance evolution not only underscore the complexity of glacier responses to climate change in this region and highlight the critical importance of continuous, high-time-resolution monitoring of glacier surface -mass changes using cameras.

Glaciers in the Muztagh Ata have been in balance or less mass loss since at least the 1970s (Kääb et al., 2015; Bolch et al., 2017, 2019a; Brun et al., 2017). Based on the high-resolution Pleiades sterio-images, Falaschi et al (2023) analysis the annual and seasonal glacier mass balance in Muztagh Ata and address that the mean mass loss of 2020, 2021 and 2022 was -0.19±0.14 m w.e., +0.15±0.27 m w.e, +0.17±0.22 m w.e. Our ground measurements at the Kangxiwa Glacier in the Muztagh Ata regions at the 2019/2020 and 2020/2021 hydrological year are consistent with the previous knowledge of less mass loss in this region. However, the mass balance in 2021/2022 and 2022/2023 hydrological years displayed the significant mass loss. In particular, the mass loss in 2022 was the most negative and agrees well with the reported unprecedented mass loss of the Urumqi No.1 Glacier in the eastern Tien Shan (Xu et al., 2024) and the notable mass loss in the European Alps (Berthier et al., 2023; Cremona et al., 2023; Voordendag et al., 2023). Similar to the Swiss glaciers, the equilibrium line altitude of the Kangxiwa Glacier rose above the glacier summits in 2022 due to heatwaves. Extremely high air temperatures covered Eurasia and North America, with long-lasting extreme heat events affecting China (Lu et al., 2023). During the summer of







2022, three heatwaves within a 40-day period caused a mass loss equivalent to over 90% of the total ablation losses for the season, demonstrating the pivotal role of heatwaves in driving local glacier ablation in this region. Similar regional heatwaves around the world (Oliver et al., 2018; Perkins-Kirkpatrick et al., 2020) pose substantial threat to mountain glaciers. Such significant mass loss by short-time heatwaves could influence the long-term dynamics of glacier ablation.

# 5.2 The anomalous circulations in July 2022 and its influence on glacier balance

The extraordinary mass loss of the Kangxiwa Glacier in the eastern Pamir and Urumqi Glacier No. 1 in the Eastern Tien Shan in 2022 (Xu et al., 2024) suggests that heatwaves may affect glacier conditions in the region on a wider scale. Analysis using ERA reanalysis data indicates that the center of the July heatwave event was located on the Pamir Plateau (Fig. 11a). The overall impact of the 2022 heatwave was felt in the western Kunlun Mountains and the Pamir Plateau — regions where glaciers were previously considered stable or in positive equilibrium (Brun et al., 2017; Kääb et al., 2015; Hugonnet et al., 2021).

We analyzed anomalies in temperature, solar radiation, cloud cover, geopotential height and wind fields at 200 hPa during July 2022, compared to the climatological average from 1970 to 2022, to investigate the anomalous large-scale atmospheric circulations linked to the 2022 July mass balance anomaly in this study area. Anomalously high temperatures of over 4°C were observed across the Pamir Plateau and the western Kunlun Mountains. This suggests that other glaciers in the region may also have experienced significant mass loss during the summer of 2022. Numerous studies have linked heatwaves and high-temperature events to anticyclonic systems (Deng et al., 2023; Jiang et al., 2023; Song et al., 2024). Figure 11b shows that an upper-level anticyclonic anomaly developed over the northwestern flank of the Tibetan Plateau at 200 hPa. Under these circulation patterns, strong easterly wind anomalies occur on the southern side of the anticyclonic anomaly, reducing the westerly wind at 30-40°N. A decrease in westerly winds means less atmospheric water is delivered to the Pamir, resulting in reduced precipitation and cloud cover (see Table 1). Abnormal anticyclonic circulation can suppress convection and cause strong subsidence, resulting in reduced cloud cover (Figure 11d). The decrease in cloud cover, in turn, leads to increased incoming shortwave radiation. Both Table 1 and ERA5 data (Figure 11c) indicate anomalously high shortwave radiation during the July 2022 heatwave events. For example, Table 1 shows that shortwave incoming radiation exceeded the average for 2020–2021–2023 by 90.6 W/m<sup>2</sup> (+31%) and 48.7 W/m<sup>2</sup> (+17%) during the two heatwave events. This enhanced solar radiation further amplified the heatwave. For example, Table 1 shows that the shortwave incoming radiation during the two heatwaves exceeded the average for 2020–2021–2023 by 90.6 W/m<sup>2</sup> (+31%) and 48.7 W/m<sup>2</sup> (+17%) respectively. This enhanced solar radiation further intensified the heatwaves. Overall, anticyclonic circulation anomalies contributed to air descent and anomalous diabatical heating, resulting in sunny and dry weather in eastern Pamir in July 2022 (Fig. 8). This favoured an increase in shortwave radiation to heat the land, which can increase outgoing longwave radiation and turbulent heat fluxes from the land surface to the atmosphere, causing higher air temperatures.


Additionally, soil moisture anomalies and teleconnections via Rossby wave trains are also potential contributing factors. Low soil moisture anomalies on the Tibetan Plateau may have intensified the development of heatwaves through land–atmosphere feedback mechanisms (Jiang et al., 2023). In the summer of 2022, heatwaves swept across many parts of the Northern Hemisphere, causing extreme heat events in North America, Europe, and the Yangtze River in China (Lu et al., 2022). The anomalous anticyclone in east Pamir appears to be linked to the propagation of wave energy from an upstream mid-latitude wave train originating in the northwestern North Atlantic (Deng et al., 2023; Song et al., 2024). Anticyclone developed above western Europe (Fig. 11b) may have acted as a conduit for Rossby wave propagation to the anticyclone developed under the Northwestern flank of TP, linking glacier mass balance in this region to large-scale climate mode such as the Atlantic Multidecadal Oscillation.

**Figure 11.** The anomalies of meteorological variables and atmospheric circulations in July 2022. (a) Air temperature; (b) Geopotential height (unit: gpm) and wind field anomalies at 200 hpa; (c) Shortwave incoming radiation (SWin, unit: W/m²) anomaly; (d) Cloud cover fraction anomaly.

# 6. Conclusions


Our study demonstrates that integrating time-lapse camera imagery with in situ glaciological measurements provides a reliable method of determining daily surface mass balance of glaciers in the east Pamir region. The results revealed the pronounced interannual variability in the surface mass balance of Kangxiwa Glacier during the 2019/2020-2022/2023

hydrological years. High temporal resolution observations enable the quantification of accumulation-ablation cycles, generally characterizing regional glaciers as "spring accumulation-summer ablation" type. Interannual variability in surface mass balance is predominantly modulated by temperature-driven ablation in summer. Notably, Kangxiwa Glacier exhibited extreme sensitivity to heatwaves during the ablation season. Three heatwaves in July–August 2022 induced over 800 mm w.e. of mass loss within 40 days. This exceptional melting brought forward Glacier Loss Day by one month, depleted spring net accumulation and caused the equilibrium line altitude to rise above the glacier's highest elevation. The 2022 event was linked to weakened westerly circulation, a large-scale atmospheric pattern affecting Central Asia. Given the recent warming and increasing frequency of the extreme weather in East Pamir, it is possible that glaciers are transitioning from a state of near-equilibrium or weak negative balance to a state of persistent mass deficit.

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

# Acknowledgement

The study was supported by the National Natural Science Foundation of China (Grant No. 422711382), Excellent Research Group Program for Tibetan Plateau Earth System (No. 42588201), Lhasa Science and Technology Plan Project (LSKJ202406), the National Key R&D Program of China (Grant No. 2024YFF0808601), Science and Technology Plan Projects of Tibet Autonomous Region (XZ202301ZY0022G), Key Innovation Team of National Climate Centre "Climate Change Monitoring and Projection in the Third Pole Region" (NCCCXTD007).

# Code and data availability

Data in this study are available upon request from the corresponding author.

#### **Author contributions**

X.Y., W.Y analysed the data and write the manuscript. M.Z.,Y.F., S.Y., P.W., W.Z. assisted in collecting all data and discussion. Funding acquisition, W.Y., B.X. All authors have read and agreed to the published version of the manuscript.

# **Competing interests**

The authors declare that they have no conflict of interests.

**Disclaimer.** Publisher's note: Copernicus Publications remains neutral with regard to jurisdictional claims in published maps and institutional affiliations.