# Peer review of "Glacier Mass Balance and Its Response to Heatwaves for Kangxiwa Glacier in the Eastern Pamir: Insights from Time-Lapse Photography and In-situ Measurements"

_EGUsphere, 2025_

## Referee Comment (RC1)

Review of: Glacier Mass Balance and Its Response to Heatwaves for Kangxiwa Glacier in the Eastern Pamir: Insights from Time-Lapse Photography and In-situ Measurements

By: Ying Xie, Baiqing Xu, Meilin Zhu, Yu Fan, Pengling Wang, Song Yang, Wenqing Zhao, Wei Yang

**20 October 2025**

The manuscript by Xie et al aims to investigate how glaciers in the eastern Pamir, specifically the Kangxiwa Glacier, respond to short-term climatic events such as heatwaves, and to better understand the processes driving glacier mass balance variability in this region. To do so, time lapse photography, in situ measurements and ERA5 data are used.

The manuscript is interesting and timely, and applies methods recently used in the European Alps to a different region (Tibetan Plateau). Before the manuscript is ready for publication, I see a few shortcomings and points that can strengthen the manuscript.

First, no clear distinction is made between methods and results. The results of the time lapse cameras are presented in the methods. This needs to be clearly separated.

Second, it is not clear how the mass balance from three time-lapse stakes is interpolated to the daily glacier-wide mass balance (Fig. 6). This needs to be clarified in the method section. This would strengthen the use of the time-lapse photography, which is presented poorly now.

Last, the manuscript (and supplementary material) contains a lot of figures, that are not all meaningful. For example, Figure 5 is Figure 4 with a factor of snow/ice density, and Figure 7 is Figure 6, but at monthly intervals. Also, Figure 10 is a different representation of Figures 4-7, without showing new data. I will comment more on this in the specific comments.

**Specific comments:**

- 1. Title: consider adding the ERA5 reanalysis data, as this is also a big contribution to the manuscript.
- 2. L43: Kunlun, the Pamir and Karakoram ranges are a part of the Tibetan Plateau, right? Please state this in the text.
- 3. L45/46: the references used here are not correct. References to the glaciological mass balance method can be: Kaser et al, 2003, and Cogley et al, 2011
- 4. L47: glacier dynamics is used, but glacier mass changes are meant. Dynamics has to do with the movement of the glacier. Replace dynamics with glacier mass changes here and throughout the paper.
- 5. L48: in this paragraph, it is not clear if it is about the Tibetan Plateau or extreme events worldwide. Please clarify.
- 6. L54: The references here deal with marine heatwaves or regional heatwaves. Especially Oliver et al is not relevant. Consider removing and finding references relevant to mountain glaciers.
- 7. L58: consider adding 'to monitor ablation' after monitoring techniques to clarify and add that the steel wire is called SmartStake. It derives ablation by rolling the wire up.
- 8. L63-66: sharpen the scope of the study and add that also ERA5 reanalysis data are used. My expectation after reading the introduction was, that it would have a focus on the time-lapse photography method, but this expectation was not met after reading the manuscript.

- 9. L71-73: is this data representative for the Kangxiwa Glacier? Why is only the mean summer temperature mentioned?
- 10. L79: for what period is this shrinkage rate?
- 11. Figure 1: please add scale bars to Figures 1b and 1d.
- 12. L90: using Yu et al 2013 gives the impression that this is the paper that describes the well-established glaciological method. Replace by Kaser et al, 2003 and Cogley et al, 2011.
- 13. L92: add a reference to the 900 kg m-3. For example, Huss 2013 and reference therewithin.
- 14. L100: here and at several locations of the paper 'stake height' is used. I think the authors mean to say 'the melt out length of the stake', because the length of the stake itself is not changing. Correct it here and throughout the paper, e.g. also in L106.
- 15. S1: this figure is subdivided in 6 to 8 periods per camera location. How are these monitoring periods defined? Would it be possible to make one long period per location?
- 16. L139 and onwards: this paragraph is already part of the results section.
- 17. L40: for the reader, it is not directly clear what the difference is between in situ stake observations and manually calculated results. Please clarify.
- 18. L142/143: these results seem surprisingly low to me, given the roughness of the glacier surface and the possibility to read stakes manually. Please elaborate.
- 19. L144: broader applications? Multiple elevations? What do you mean? Please give examples.
- 20. Figure 4: What is the white period in this figure? Add this to the caption and the text. Also, the figure is confusing, given negative values above positive values on the y-axis. Consider renaming 'changes in stake height' and mirror the y-axis. Please also explain what happened in the last 2 months of 2022/2023 at location 5005 m. There is a lot of data missing there. In the caption replace 'the stake height was set to 0 cm' by changes relative to the first day of the hydrological year (otherwise positive values are not possible).
- 21. L152: how are the surface conditions derived? In S2, there is only distinguished between snow and ice, whereas later (L161) also firn is mentioned.
- 22. L154: why is the average snow density from June used? Is the representative? What is the variability in snow density over years and over the seasons?
- 23. L156: So Figure 5 is derived with interpolated data from the 10 stake locations in Figure 1? Or how can you have daily mass balances? This part of the method really needs to be clarified.
- 24. L157: which differences do you mean? The differences between the camera observations and the in situ measured mass balance? How is this derived? And with 3 points, are the three locations of the cameras meant?
- 25. L158: I find those standard deviations relatively high and it is concluded that the camera approach in reliable. Please elaborate on this.
- 26. L158-163: Why is this text in italics? Again, in this section, it needs to be made clear how you distinguish between snow, ice and firn and how this is derived from the images.
- 27. L165: the DEM with this resolution is derived from which satellite (mission). What is the temporal resolution of the DEMs? In this paragraph, it is also not clear how and why the DEMs are used. Later, I don't see any results that are derived from the DEMs.
- 28. L170/171: confusion is added, as here it is referred to the cameras again. In which figure can I find results? What does it mean?
- 29. Figure 5: shorten the extent of the y-axis for the upper two locations. Consider adding all locations to one plot or display it in a similar way as figure 4, grouped over years. This figure also raised the question how the cameras work in winter, do they get snowed in?
- 30. L178: The AWS was deployed since 2011, but does it cover the entire period until 2023? Are these data representative for the Kangxiwa Glacier?

- 31. L185-188: elaborate more on how and why ERA5 data is used. This should already have been mentioned in the scope/aim of the study.
- 32. L191: it is unclear from the text how the glacier-wide mass balance is calculated at such high temporal resolution. Are only the three camera stakes used or also the other ablations stakes, as indicated in Fig. 1d? I cannot imagine it's sufficient to interpolate only from the three stakes. If this is the case, please elaborate why this would be sufficient.
- 33. L196/197: authors claim they have evidence for sublimation or mechanical snow erosion, but abrupt transition of the surface cover is no evidence for this, especially because it's not related to wind velocity and direction and/or temperature. Additionally, it is not clear how the surface properties were derived, so calling it abrupt is not appropriate here.
- 34. L198: substantial, please quantify this.
- 35. Figure 6: I recommend to make three subplots here in total. a) Figure of the mass balance of all the years, b) cumulative precipitation over all the years in order to get a feeling of the total amount of precipitation over the year (so not the daily bars that are hard to compare), c) CPPD, as it the current Fig. 6e now.
- 36. L213: from fig 6a, there is no GLD visible in 2019/2020. The fact that a GLD is given in the text, makes me doubt even more about the reliability of (the interpolation of) the glacierwide mass balance.
- 37. L214: please also relate the occurrence of the Glacier Loss Day to the total amount of precipitation, like it has been done in the original GLD paper (Voordendag et al., 2023).
- 38. Figure 7: this figure is unnecessary in my opinion. Especially if adaptations are made to Fig 6 (see comment 36).
- 39. L229-231: this line would fit better in the discussion section.
- 40. L235: "unprecedented" has been used as strong statement by Cremona et al. (2023), but this statement seems too strong here, as we do not see any proof how it has been before. We do not have a long period of observations at Kangxiwa Glacier.
- 41. L236: from Fig. 3c it is not clear to me that the warm summer of 2022 was unprecedented. Please rephrase, for example, "xx % of the time above the 90% percentile".
- 42. L239: do you refer to air temperature data here? Please clarify.
- 43. S3: is the slope in Fig. S3b in °C/year? Also add "point" to grid in the caption and correct the typo EAR-land.
- 44. Figure 8: taking an average of 4 years is not very helpful. Is 2022 removed from this average. I suggest plotting all the years in one subplot, without averaging. Also, where are the three heat events based on? What was the threshold (temperature?)? Furthermore, I wonder how representative the location of this AWS is for Kangxiwa Glacier. Please elaborate.
- 45. Table 1: calculating anomalies relative to only three other years is not sufficient. To give information about the climate, a longer period is needed. Consider removing Table 1.
- 46. Figure 10: this figure is unnecessary. It is visually hard to interpret, and the data has already been displayed in the previous figures. Precipitation and CPDD are not related variables, and it is thus not relevant to plot them against each other.
- 47. L293: the reference to Maussion et al 2012 is not found in the reference list. I might also have missed other references. Please check this!
- 48. L302-305: how is this comparative analysis calculated? What is meant with accumulation variability accounts for ~33%? I'd recommend relating the maximum of precipitation and the total ablation to the total mass loss. "Variability in mass loss" is not a clear indicator. Also, refrain from using italics here.
- 49. L316: less mass loss? Less than? Please rephrase. I think a near-equilibrium state is meant.
- 50. L320: less mass loss? See 49.

- 51. L329: the reference to Oliver et al. (2018) is about marine heatwaves and not relevant here. See comment 6.
- 52. L334: what is a 'center of a heatwave'?
- 53. L336: an equilibrium cannot be positive. Rephrase to: 'glaciers were previously considered in equilibrium or having a positive glacier mass balance', or similar.
- 54. L358: what are Rossby wave trains? Please add references to this statement.
- 55. L360-362: this statement is more appropriate for the introduction.

**Grammar and typos:**

- 56. L18: remove 'the' before heatwaves
- 57. L20: typo 2019/2000
- 58. L20: consider using 'and' instead of 'but'. When using 'but', you would expect mass gain for the other 2 years.
- 59. L23: heatwaves
- 60. L24: Glacier Loss Day, remove mass
- 61. L25: wakened → weakened
- 62. L27: remove 'the' before heatwaves
- 63. L32: occurred → observed
- 64. L34: Hewiit → Hewitt
- 65. L35: maybe be → might
- 66. L37: are 'a' critical component
- 67. L41: 'the' Tibetan Plateau
- 68. L55: Traditional glaciological method and geodetic survey → The glaciological method and geodetic mass balance method
- 69. L56: dynamics → differences
- 70. L60: add 'and' before terrestrial laser scanning
- 71. L61: remove 'the' before extreme melt events
- 72. L63: developed → introduced
- 73. L106, 107, 144, 148-151, 195: hight  $\rightarrow$  height, but this actually needs to be the melt out length of the stake.
- 74. L118: S channel → S-channel
- 75. L145: dynamics → changes.
- 76. L158: glacial → glacier
- 77. L177: Fig 1 → fig 1b
- 78. L207: "which own significant mass loss". Please clarify/rewrite.
- 79. L239: the corresponding grid → add "point"
- 80. L252: add a reference to Fig. 8b after the radiation values.
- 81. L317: analysis → analyses
- 82. L319: remove ground.
- 83. L320: At a regions  $\rightarrow$  region.
- 84. L328: remove the extra point after region.
- 85. L330: replace dynamics with response.
- 86. L352-354: repetition of the previous sentence. Also, see comment 45.
- 87. L365: abbreviation TP is not introduced before. Tibetan Plateau, right?
- 88. L376: "spring accumulation-summer ablation". Check spelling with hyphens here and make it the same throughout the paper. Is this type invented by the authors or have other glaciers similar behaviour?

- 89. L379: brought forward? Please rephrase. For example, the GLD occurred one month earlier in 2022 compared to the other three years, due to exceptional melting early in the season and/or under average precipitation amounts in winter.
- 90. L383: replace deficit by loss.

**References:**

- Kaser G, Fountain A, Jansson P et al. (2003) A manual for monitoring the mass balance of mountain glaciers. UNESCO, Paris
- Cogley JG, Hock R, Rasmussen L, Arendt A, Bauder A, Braithwaite R, Jansson P, Kaser G, Möller M, Nicholson L and Zemp M (2011) Glossary of glacier mass balance and related terms. IHP-VII technical documents in hydrology, 86
- Huss M (2013) Density assumptions for converting geodetic glacier volume change to mass change. *The Cryosphere*, 7(3), 877-887 (doi: 10.5194/tc-7-877-2013)

---

## Author Comment (AC1)

**Response to RC1**

The manuscript by Xie et al aims to investigate how glaciers in the eastern Pamir, specifically the Kangxiwa Glacier, respond to short-term climatic events such as heatwaves, and to better understand the processes driving glacier mass balance variability in this region. To do so, time lapse photography, in situ measurements and ERA5 data are used. The manuscript is interesting and timely, and applies methods recently used in the European Alps to a different region (Tibetan Plateau).

Response: We thank the anonymous reviewer for his/her insightful comments and constructive suggestions. Below are our point-by-point responses in blue.

**Major comments:**

Before the manuscript is ready for publication, I see a few shortcomings and points that can strengthen the manuscript. First, no clear distinction is made between methods and results. The results of the time lapse cameras are presented in the methods. This needs to be clearly separated.

Response: Following your suggestion, we restructured both the method and results Sections. We have removed all the results of the time lapse cameras, which was presented in the methods in previous version to the Results section in the revised manuscript. And we also added new Section 4.1: The performance of glacier mass balance estimation based on the time-lapse cameras in the Result Section. In this section, we compared the performance of mass balance calculated by both stake method and camera-based method in both point scale and glacier-wide scale.

Second, it is not clear how the mass balance from three time-lapse stakes is interpolated to the daily glacier-wide mass balance (Fig. 6). This needs to be clarified in the method section. This would strengthen the use of the time-lapse photography, which is presented poorly now.

Response: Thanks for pointing out such issues. Glacier-wide mass balance was derived via a weighted average of the three pointing mass balance observations by Cameras and their representative elevation-area. Although there are only three cameras available, their spatial representativeness was carefully considered before the installation near the terminus (5005m asl), middle (5137m asl) and upper region (5300 m asl) of Kangxiwa Glacier. The location of camera 1 at 5005m roughly represents the zone of $5005\pm60$ m in elevations (~4960-5080m asl); Camera2 represents the zonal range of $5137\pm60$ m in elevations (~5080 to ~5200 m asl); Camera3 represents the zonal range of $5300\pm100$ m in elevations (~5200 to ~5390 m asl). Actually, the topography condition of Kangxiwa Glacier is relatively flat and the 10 stake mass balance measurement evidenced that there are linear mass balance gradients (See the following Figure). It allows us to use the weighing method to calculate the glacier-wide mass balance.

[Figure]

Figure 1R The elevation distribution of point mass balances measured by 10 stakes over 2019/2020-2022/2023 hydrologic years

And we also compared the glacier-wide winter/summer mass balance calculated by using the traditional in-situ ablation method and the time-lapse photography on the Kangxiwa Glacier. The mean difference is within 62 mm w.e. and such comparison evidenced the reliability of the area-weight method by using three mass balance observations. We also added the relevant description to address the process in the Method section and their reliability of both point and wide-glacier mass balance in Result section in the revised manuscript (Figure 4).

[Figure]

**Figure 4** The comparison of pointed mass balance (a) and glacier-wide mass balance (b) measured by using the stake method and the time-lapse photography on the Kangxiwa Glacier.

Last, the manuscript (and supplementary material) contains a lot of figures, that are not all meaningful. For example, Figure 5 is Figure 4 with a factor of snow/ice density, and Figure 7 is Figure 6, but at monthly intervals. Also, Figure 10 is a different representation of Figures 4-7, without showing new data. I will comment more on this in the specific comments.

Response: We have deleted the redundant Figures and new Figures are provided to show the performance of daily mass balance estimation by using time-lapse camera (New Figure 4), the climatic backgrounds from long-periods (New Figure 6). Please see the revised Figures and descriptions.

**Specific comments:**

1.    Title: consider adding the ERA5 reanalysis data, as this is also a big contribution to the manuscript.

Response: The ERA5 reanalysis temperature data is important to this research. We think the camara mass balance data are the core data to the manuscript. We did not therefore add "the ERA5 reanalysis data" in the title. However, following your specific comment 8 and comment 31 and RC2's comment, we further addressed the role of ERA5 reanalysis data in both objectives and methods and the title was revised as "Glacier Mass Balance and Its Response to 2022 Heatwaves for Kangxiwa Glacier in the Eastern Pamir: Insights from Time-Lapse Photography".

2.    L43: Kunlun, the Pamir and Karakoram ranges are a part of the Tibetan Plateau, right? Please state this in the text.

Response: The sentence was revised as: *In addition, long-term glaciological measurements are scarce across the western Tibet Plateau, particularly in the western Kunlun, the Pamir and Karakoram ranges (Barandun and Pohl, 2023; Yao et al., 2022; Zemp et al, 2023).*

3.    L45/46: the references used here are not correct. References to the glaciological mass balance method can be: Kaser et al, 2003, and Cogley et al, 2011

Response: Thanks for the comment. We changed the reference here and add them to the References.

Kaser G, Fountain A, Jansson P et al. (2003) A manual for monitoring the mass balance of mountain glaciers. UNESCO, Paris

Cogley JG, Hock R, Rasmussen L, Arendt A, Bauder A, Braithwaite R, Jansson P, Kaser G, Möller M, Nicholson L and Zemp M (2011) Glossary of glacier mass balance and related terms. IHP-VII technical documents in hydrology, 86

4.    L47: glacier dynamics is used, but glacier mass changes are meant. Dynamics has to do with the movement of the glacier. Replace dynamics with glacier mass changes here and throughout the paper.

Response: We used "glacier mass changes" here and we have unified throughout the paper.

5. L48: in this paragraph, it is not clear if it is about the Tibetan Plateau or extreme events worldwide. Please clarify.

Response: This paragraph is about extreme events worldwide. The sentence was revised as: *Recent extreme events such as heatwaves have caused abnormal high-elevation melting in the world.*

6. L54: The references here deal with marine heatwaves or regional heatwaves. Especially Oliver et al is not relevant. Consider removing and finding references relevant to mountain glaciers.

Response: We have removed Olive et al. in the introduction and discussion. We added two references on heatwave impact on mountain glaciers in Apls and Tibetan Plateau.

Colucci R., Giorgi F., Torma C., Unprecedented heat wave in December 2015 and potential for winter glacier ablation in the eastern Alps, Scientific Report, 7: 7090, DOI:10.1038/s41598-017-07415-1, 2017.

Zhu, F., Zhu, M., Yang, W., Wang, Z., Guo, Y., Yao, T., Drivers of the extreme early spring glacier melt of 2022 on the central Tibetan Plateau. Earth and Space Science, 11, e2023EA003297. https://doi.org/10.1029/2023EA003297, 2024.

7. L58: consider adding 'to monitor ablation' after monitoring techniques to clarify and add that the steel wire is called SmartStake. It derives ablation by rolling the wire up.

Response: The sentence was revised as "*Recent advancements in high-temporal-resolution techniques to monitor ablation, such as the SmartStake deriving the ablation by rolling the steel wire up (A2PS contributors 2021), automated cameras monitoring colour-coded ablation stakes (Landmann et al., 2021; Cremona et al., 2023), and terrestrial laser scanning techniques (Voordendag et al., 2023), have provided new insights into short-term mass balance variations*"

8. L63-66: sharpen the scope of the study and add that also ERA5 reanalysis data are used. My expectation after reading the introduction was, that it would have a focus on the time-lapse photography method, but this expectation was not met after reading the manuscript.

Response: The sentence was changed as: "*Based on these high-resolution daily surface mass balance datasets, this study aims to (1) characterize the contrasting seasonal mass balance patterns of the Kangxiwa Glacier in the eastern Pamir under varying climatic forcing regimes across the 2019/2020 to 2022/2023 balance years; (2) quantify the sensitivity of surface mass balance to the extreme 2022 summer heatwaves; and (3) identify the atmospheric circulation anomalies linked to the 2022 heatwaves using ERA5 reanalysis data, and further elucidate regional glacier response.*"

9. L71-73: is this data representative for the Kangxiwa Glacier? Why is only the mean summer temperature mentioned?

Response: This data was used here to illustrate the regional climate background (Figure 1b), showing that the Kangxiwa Glacier is situated in a cold and dry climate. Given that mean summer temperature was inappropriate for this purpose, it was replaced with mean annual temperature. Also, references about the values were added here following RC2's comment The sentence was changed to "*Data from the Taxkorgan Meteorological Station (3091 m a.s.l., ~50 km south of Muztagh Ata) show a mean annual temperature of 3.7°C and annual precipitation of ~70 mm (Li et al., 2022; Lv et al., 2020).*"

Li Z., Wang N., Chen A., Liang Q., and Yang D., (2022) Slight change of glaciers in the Pamir over the period 2000–2017, Arctic, Antarctic, and Alpine Research, 54(1), 13-24, DOI: 10.1080/15230430.2022.2028475.

Lv M, Quincey DJ, Guo H, King O, Liu G, Yan S, Lu X, Ruan Z (2020). Examining geodetic glacier mass balance in the eastern Pamir transition zone. Journal of Glaciology, 66 (260), 927–937. https://doi.org/ 10.1017/jog.2020.54

10. L79: for what period is this shrinkage rate?

Response: The period of 2000-2017 was added it in the revised manuscript.

11. Figure 1: please add scale bars to Figures 1b and 1d.

Response: We added scale bars to Figure 1b and 1d. The symbol of the studied was changed to triangle.

[Figure]

**Figure 1.** Study region and the distribution of in-situ measurements. (a) Location of the studied area in the eastern Pamir, central Asia; (b) The location of Kangxiwa Glacier (red triangle), Automatic Weather Station (AWS) at the elevation of 4900 m and Taxkorgan meteorologic station (red stars); the World hillshade layer was used as the background (http://www.arcgis.com) and the outline of the glacier was obtained from the Randolph glacier inventory (RGI 7.0; http://www.glims.org/RGI); (c) Photograph of the time-lapse camera and colour-coded stake on the Kangxiwa Glacier (5300 m a.s.l., 20 June, 2020); (d) Topographic map of the Kangxiwa Glacier showing the locations of three time-lapse camera monitoring systems and the ablation stakes/snow pits for in-situ mass balance observations.

12. L90: using Yu et al 2013 gives the impression that this is the paper that describes the well-established glaciological method. Replace by Kaser et al, 2003 and Cogley et al, 2011.

Response: Thanks for the suggestion. We changed the references, added Kaser et al, 2003 and Cogley et al, 2011 to the reference part.

13. L92: add a reference to the 900 kg m-3. For example, Huss 2013 and reference therewithin.

Response: We added the reference.

14. L100: here and at several locations of the paper 'stake height' is used. I think the authors mean to say 'the melt out length of the stake', because the length of the stake itself is not changing. Correct it here and throughout the

paper, e.g. also in L106.

Response: Following your suggestion, we used "the stake melt-out length" in the revised manuscript.

15. S1: this figure is subdivided in 6 to 8 periods per camera location. How are these monitoring periods defined? Would it be possible to make one long period per location?

Response: These monitoring periods at each site were founded on the date of the glacier field investigation. For example, "5005: June 16, 2022–September 25, 2022" indicates that glacier mass balance observations were conducted on June 16 and September 25, 2022 at the 5005 observational site. We replotted all the data at each location as follow.

[Figure]

**Figure S1** Linear relationship between $L_p$ and L for each monitoring period at each camera site.

16. L139 and onwards: this paragraph is already part of the results section.

Response: Figure 3 was removed to the supplementary material. Figure 4 was deleted according to the major comment.

17. L140: for the reader, it is not directly clear what the difference is between in situ stake observations and manually calculated results. Please clarify.

Response: For clarify, we revised the terms to "field tape measurements" and the "manually photo inspection". We also replotted the Figure 3 for more clarification, and moved it to the supplementary material.

[Figure]

**Figure S2** The performance of stake melt-out lengths derived automatically from time-lapse cameras (black line), compared with in-situ observations (red square) and manual inspections (blue triangle) for a stake at the elevation of 5005 m asl. Noted that the camera monitoring systems were maintained in June and September in each year during the period from 2020 to 2023.

18. L142/143: these results seem surprisingly low to me, given the roughness of the glacier surface and the

possibility to read stakes manually. Please elaborate.

Response: We have deleted the relevant description, and only provided the value ranges with centimeter-level precision.

19. L144: broader applications? Multiple elevations? What do you mean? Please give examples.

Response: We did not mention the broader application in the revised manuscript. We only address the robustness of the semi-automatic procedure for deriving stake length changes by time-lapse cameras.

20. Figure 4: What is the white period in this figure? Add this to the caption and the text. Also, the figure is confusing, given negative values above positive values on the y-axis. Consider renaming 'changes in stake height' and mirror the y-axis. Please also explain what happened in the last 2 months of 2022/2023 at location 5005 m. There is a lot of data missing there. In the caption replace 'the stake height was set to 0 cm' by changes relative to the first day of the hydrological year (otherwise positive values are not possible).

Respond: Thanks for point out the incorrect description in figure caption. Regarding the redundant Figures, this Figure has been deleted according to Major comment and another review's comment. The white period was the missing data. Missing data during the last 2 months of 2022/2023 at the 5005 m was caused by that the top part of the stake was out of the camera view. "changes in stake height" was revised to "Stake melt-out length" (comment 14) in figures in the revised version.

21. L152: how are the surface conditions derived? In S2, there is only distinguished between snow and ice, whereas later (L161) also firn is mentioned.

Respond: The surface conditions (ice/snow) were determined via visual interpretation of the captured images. Given the challenge associated with differentiating between snow and firn, we no longer make this distinction in the revised manuscript. We have added the relevant description "*The daily changes in the stake melt-out length recorded by three cameras were converted into changes in mass balance by multiplying by the corresponding density for different surface conditions (snow vs. bare ice) through visual inspection of the photos manually.*"

22. L154: why is the average snow density from June used? Is the representative? What is the variability in snow density over years and over the seasons?

Respond: The density of snow pits was measured during each field mass balance observation on June and September. The average density of snow pits at the three sites in June was 426 kg/m$^3$, ranging from 286 kg/m$^3$ to 587 kg/m$^3$; while that in September was 380 kg/m$^3$, ranging from 295 kg/m$^3$ to 536 kg/m$^3$. The average density from all the snow pits in June and September was 405±69 kg/m$^3$. In the revised version, we used 405 kg/m$^3$ as the snow density for calculation.

23. L156: So Figure 5 is derived with interpolated data from the 10 stake locations in Figure 1? Or how can you have daily mass balances? This part of the method really needs to be clarified.

Response: The data in Figure 5 is derived from three camera locations at 5005 m a.s.l., 5137 m a.s.l. and 5300 m

a.s.l., respectively. Daily balance was calculated as the product of the daily change in the stake melt-out length and the corresponding density. As response to major Comment 2, we have clarified the description of how glacier-wide mass balance was calculated and the new Figure 4 was provided to show their reliability by comparing with the traditional stake-based glacier-wide mass balance.

24. L157: which differences do you mean? The differences between the camera observations and the in situ measured mass balance? How is this derived? And with 3 points, are the three locations of the cameras meant?

Response: The 3 cameras locations (5005 m, 5137 m and 5300 m) also have the coupling in-situ stake and snow pit measurements (close to the camera). To validate the accuracy of camera-based mass balance observation, we compare the camera-based and stake-based mass balance at three locations. For clarify, this sentence was revised as "*Quantitative comparisons reveal that the mean seasonal mass balance differences between the two datasets at the three monitoring sits are −64, −13, and 2 mm w.e, with corresponding standard deviations of 193, 137 and 104 mm w.e., respectively*"

25. L158: I find those standard deviations relatively high and it is concluded that the camera approach in reliable. Please elaborate on this.

Response: We actually compared the winter and summer mass balance estimated by between camera-based and stake-based methods. Please see the following figure. The standard deviation is relatively small (seasonal average difference was 26±149) if comparing with winter and summer mass balance at different locations. Following your suggestion, we added the relevant description of summer and winter mass balance in the revised manuscript.

[Figure]

Figure 5a. The comparison of pointed mass balance measured by using the stake method and the time-lapse photography on the Kangxiwa Glacier.

26. L158-163: Why is this text in italics? Again, in this section, it needs to be made clear how you distinguish between snow, ice and firn and how this is derived from the images.

Response: Sorry for the type error. In this manuscript, the glacier surface conditions are divided into two categories:

ice and snow. We manually distinguished between ice and snow by visually inspecting the photos. We added this '*The daily changes in the stake melt-out length recorded by three cameras were converted into changes in mass balance by multiplying by the corresponding density for different surface conditions (snow vs. bare ice) through visual inspection of the photos manually*'

27.  L165: the DEM with this resolution is derived from which satellite (mission). What is the temporal resolution of the DEMs? In this paragraph, it is also not clear how and why the DEMs are used. Later, I don't see any results that are derived from the DEMs.

Response: Using the 30 m resolution SRTM DEM, the glacier area at different elevation bans were generated to calculate the glacier-wide mass balance by using camera-based and stake-based method. We add this information in the text as "*A 30-m resolution SRTM DEM was employed to quantify the area distribution with its elevations.*"

28.  L170/171: confusion is added, as here it is referred to the cameras again. In which figure can I find results? What does it mean?

Response: L170/L171 was used to describe how to fill the data gaps which were referring to new Figure 3.

29.  Figure 5: shorten the extent of the y-axis for the upper two locations. Consider adding all locations to one plot or display it in a similar way as figure 4, grouped over years. This figure also raised the question how the cameras work in winter, do they get snowed in?

Response: We have replotted the Figure 5 following your suggestion. The camera work in winter continuously and thus the changes in stake melt-out length could be detected.

[Figure]

Figure 3. Comparison of the accumulated mass balance estimated using time-lapse cameras (lines) and the glaciological methods (triangles) at the three locations (5005 m, 5137 m and 5300 m) on the Kangxiwa Glacier. The thin dotted lines denote gap-filled data.

30. L178: The AWS was deployed since 2011, but does it cover the entire period until 2023? Are these data representative for the Kangxiwa Glacier?

Response: The 4900AWS data started from Mid-July 2011 and cover the entire period until 2023. This AWS is the only long-term continuous measurement near the Kangxiwa Glacier in high elevation. We thus use this data to investigate their climatic background in the following section.

31. L185-188: elaborate more on how and why ERA5 data is used. This should already have been mentioned in the scope/aim of the study.

Response: We have elaborated the description the usage of ERA data in the revised manuscript. "*To investigate the possible climate mechanism on abnormal glacier mass loss, this study also employed the fifth generation of reanalysis data from the European Centre for Medium-Range Weather Forecasts (ERA5) with 2.5° horizontal resolution, which has been widely used in climate research (Hoffmann et al., 2019; Song et al., 2024) and glacier change analyses (Zhu et al., 2024b). By analysing the spatial anomalies pattern of geopotential height, wind fields, and surface air temperature, we investigated the change of large-scale atmospheric circulation and discussed its possible influences on extreme weather events and the subsequent responses of glacier surface mass balance.*"

32. L191: it is unclear from the text how the glacier-wide mass balance is calculated at such high temporal resolution. Are only the three camera stakes used or also the other ablations stakes, as indicated in Fig. 1d? I cannot imagine it's sufficient to interpolate only from the three stakes. If this is the case, please elaborate why this would be sufficient.

Response: The glacier-wide daily mass balance was calculated based on the three camera's data. We firstly calculated the daily mass change at each point (comment 23), glacier-wide daily mass balance was then derived via a weighted average of the three-point values, with weights assigned by elevation-specific area proportions. The three measurement points represented elevation ranges of 4960–5080 m (weight: 0.33), 5080–5200 m (weight: 0.38), and 5200–5390 m (weight: 0.29). The three cameras are located at the lowest, the middle part, and the highest part of the glacier, respectively. They cover the sites with the lowest, medium, and highest surface mass balance values of the glacier. The in-situ observations at the 10 points (Figure 1d) showed strong liner relationship with elevation (Fig R1), which provides favorable conditions for calculating the mass balance of the entire glacier using the three camera's data. We also used the in-situ seasonal mass balance data to test whether it can be used the three point values to generate the glacier-wide mass change data. The difference between the seasonal three camera-based and ten-stake based mass balance values was 62±54 mm w.e.. The low difference further confirms that using the 3 point mass change values and weighted factor to drive glacier-wide mass balance was a reasonable approach.

33. L196/197: authors claim they have evidence for sublimation or mechanical snow erosion, but abrupt transition of the surface cover is no evidence for this, especially because it's not related to wind velocity and direction and/or

temperature. Additionally, it is not clear how the surface properties were derived, so calling it abrupt is not appropriate here.

Response: Thanks for pointing out this issue. The low air temperature in winter prohibits the surface melting. The time lapse camera evidenced that the snow-cover surface was transferred to exposed ice surface condition, which means the possible surface sublimation or mechanical snow drift in the winter season. Therefore, the sentence was revised as "*The slight surface mass loss during this period was likely caused by sublimation or mechanical snow drift by strong winter wind, which was evidenced by the transition from snow-covered surfaces to exposed bare ice (Fig. S2)*".

34. L198: substantial, please quantify this.

Response: We rephrased it as "*During the accumulation period (mid-April to June/July), the glacier experienced different maximum snow accumulation across the entire glacier ranged from 258 to 445 mm w.e.*"

35. Figure 6: I recommend to make three subplots here in total. a) Figure of the mass balance of all the years, b) cumulative precipitation over all the years in order to get a feeling of the total amount of precipitation over the year (so not the daily bars that are hard to compare), c) CPPD, as it the current Fig. 6e now.

Response: Following your suggestion, we have redrawn following new Figure 5.

[Figure]

**Figure 5.** Cumulative glacier-wide mass balance of the Kangxiwa Glacier during the 2019/2020–2022/2023 hydrological years with the uncertainties by shaded area (a). The cumulated precipitation (b) and the cumulated positive

36. L213: from fig 6a, there is no GLD visible in 2019/2020. The fact that a GLD is given in the text, makes me doubt even more about the reliability of (the interpolation of) the glacier-wide mass balance.

Response: Our impression is wrong. The original intention was to use "~ 20 August 2020" to mean that the 2019/2020 hydrologic year almost reach its GLD at 20 August 2020. We revised sentence as: *Concurrently, the GLD-defined as the date when the net mass balance transitions to negative and all winter snow accumulation is depleted (Voordendag et al., 2023)-occurred on around 11 July 2022, approximately one month earlier than in other years (e.g., ~3 September 2021; ~26 August 2023). Notably, no GLD was recorded in the 2019/2020 hydrologic year, a phenomenon attributed to abundant winter-spring snow accumulation coupled with relatively low summer air temperature (Fig. 5b,c).*

37. L214: please also relate the occurrence of the Glacier Loss Day to the total amount of precipitation, like it has been done in the original GLD paper (Voordendag et al., 2023).

Response: We compared the cumulative precipitation before GLD between these hydrologic years, and analyzed the relationship between GLD and the amount of precipitation. The added sentences are: *The earliest GLD in the 2021/2022 hydrologic year was partly caused by the lowest precipitation in the accumulation season. Specifically, the AWS4900 show that the cumulative precipitation for the 2021/2022 hydrologic year was only 131.8 mm until 11 July, while the corresponding values were 195.9 mm, 165.1 mm, and 218.7 mm for the 2019/2020, 2020/2021, and 2022/2023 hydrologic years, respectively.*

38. Figure 7: this figure is unnecessary in my opinion. Especially if adaptations are made to Fig 6 (see comment 36).

Response: Following the comment, Figure 7 was moved to supplementary material.

39. L229-231: this line would fit better in the discussion section.

Response: We use this in the discussion part.

40. L235: "unprecedented" has been used as strong statement by Cremona et al. (2023), but this statement seems too strong here, as we do not see any proof how it has been before. We do not have a long period of observations at Kangxiwa Glacier.

Response: We have removed the "unprecedented" from the subtitle.

41. L236: from Fig. S3c it is not clear to me that the warm summer of 2022 was unprecedented. Please rephrase, for example, "xx % of the time above the 90% percentile".

Response: Thanks for the suggestion, we rephrased these sentences. We also added one new Figure 7 to address the anomalous high air temperature in July of 2022. "*Both ERA reanalysis data and ground-based meteorological station observations revealed an unprecedented summer warming event in 2022 in the east Pamir, with the most pronounced anomaly recorded in July (Fig. 6 and Fig. S5). The ERA5 temperature data confirmed that the corresponding grid point of the Kangxiwa Glacier in July 2022 was the highest recorded between 1981 and 2024 (Fig. 6a). The AWS4900 records evidenced that the average July temperature during 2012-2023 was 2.9°C, while it reached 6.2°C in July 2022, constituting a substantial positive temperature anomaly (Fig. 6b). Additionally, daily maximum temperature recorded at Taxkorgan station during summer 2022 was significantly higher than the long-term mean for 1957-2023 (Fig. S5), with 61% of days in July exceeding the 90th percentile of historical temperatures and satisfying the criteria for an extreme heat event as defined by Lu et al. (2024). This evidence indicates that July 2022 was characterized by significantly elevated temperatures relative to historical baselines. This exceptional climatic event provides a unique opportunity to analyse how glaciers respond to extreme heatwaves in the eastern Pamir.*"

[Figure]

**Figure 6.** Variation of mean ERA-land air temperature in July at the corresponding point grid of Kangxiwa Glacier during the period from 1981 to 2024 (a) and heatmap of representative monthly air temperature from May to September during 2012-2023 recorded by AWS4900 (b)

42. L239: do you refer to air temperature data here? Please clarify.

Response: Yes, we refer to ERA air temperature. The sentence was revised as: "*The ERA5 temperature data confirmed that the corresponding grid point of the Kangxiwa Glacier in July 2022 was the highest recorded between 1981 and 2024 (Fig. 6a).*".

43. S3: is the slope in Fig. S3b in C/year? Also add "point" to grid in the caption and correct the typo EAR-land.

Response: Thanks. We have added the units ℃/year in the legend. Also, we have added "point" to grid and correct the typo as ERA-land. According to comment 41, the subplot showing the July temperature data was removed from the original figure, and the revised version is provided below.

[Figure]

Figure S5 Variation of mean ERA-land air temperature in July (a) and July-September (b) at the corresponding grid point of Kangxiwa Glacier during the period from 1981 to 2024. (c) Comparison of June-September daily max air temperature in 2022 with the 1957-2023 mean at Taxkorgan station.

44. Figure 8: taking an average of 4 years is not very helpful. Is 2022 removed from this average. I suggest plotting all the years in one subplot, without averaging. Also, where are the three heat events based on? What was the threshold (temperature?)? Furthermore, I wonder how representative the location three heat events of this AWS is for Kangxiwa Glacier. Please elaborate.

Response: Precipitation, temperature, radioactive were the average of 2020 2021 and 2023. The three heat events were identified by comparing the air temperature of 2022 with that of 2020, 2021, and 2023 average. The threshold was determined by that the 2022 temperature was higher than that of 2020,2021and 2023 mean. We plotted the variables during the four years in one subplot without averaging (Fig. 2R). However, this figure makes it difficult for readers to distinguish the differences between the variables for 2022 and those for the other three years, so we kept the original figure. Given the short distance between the meteorological station and the glacier, the variation

trends of air temperature at the two sites should be consistent; thus, the heatwave events recorded by the AWS4900 can be used to represent the heatwave conditions experienced by the glacier.

[Figure]

Figure 2R. Comparison of (a) daily air temperature, (b) daily incoming shortwave radiation, (c) daily precipitation and (d) daily glacier-wide mass balance at the Kangxiwa Glacier during the ablation season (June–September), showing the difference between 2022 and the other three years. The grey line shows the average for 2020, 2021 and 2023 with standardized variation (grey shaded area), while the red line shows the records for 2022. Light red rectangles highlight three heat events. The meteorological data was derived from AWS4900.

45. Table 1: calculating anomalies relative to only three other years is not sufficient. To give information about the climate, a longer period is needed. Consider removing Table 1.

Response: Following your suggestion, we have removed the Table 1.

46. Figure 10: this figure is unnecessary. It is visually hard to interpret, and the data has already been displayed in the previous figures. Precipitation and CPDD are not related variables, and it is thus not relevant to plot them against each other.

Response: We have deleted the Figure.

47. L293: the reference to Maussion et al 2012 is not found in the reference list. I might also have missed other references. Please check this!

Response: We have added the missing reference and carefully recheck all references.

48. L302-305: how is this comparative analysis calculated? What is meant with accumulation variability accounts for ~33%? I'd recommend relating the maximum of precipitation and the total ablation to the total mass loss. "Variability in mass loss" is not a clear indicator. Also, refrain from using italics here.

Response: Thanks for point out this. In the revised manuscript, we rephrased it and listed the detailed numbers. "*A comparative analysis between the 2019/2020 and 2021/2022 reveals that less accumulation of 167 mm w.e. in winter season, but more ablation of 494 mm w.e. in summer season in 2021/2022. These results emphasize that interannual variability in the glacier mass balance of Kangxiwa Glacier was predominantly driven by the variability in mass loss during the ablation season.*"

49. L316: less mass loss? Less than? Please rephrase. I think a near-equilibrium state is meant.

Response: The sentence has been corrected as: *Glaciers in the Muztagh Ata have been in a near-equilibrium state since at least the 1970s.*

50. L320: less mass loss? See 49.

Response: it was revised to "near-equilibrium" here.

51. L329: the reference to Oliver et al. (2018) is about marine heatwaves and not relevant here. See comment 6.

Response: We changed the reference as Colucci et al., 2017 and Zhang et al., 2025.

Colucci R., Giorgi F., Torma C., Unprecedented heat wave in December 2015 and potential for winter glacier ablation in the eastern Alps, Scientific Report, 7: 7090, DOI:10.1038/s41598-017-07415-1, 2017.

Zhang, T., Deng, G., Liu, X., He, Y., Shen, Q., and Chen Q.: Heatwave magnitude quantization and impact factors analysis over the Tibetan Plateau. npj Clim. Atmos. Sci., 8, 2, https://doi.org/10.1038/s41612-024-00877-x, 2025.

52. L334: what is a 'center of a heatwave'?

Response: We have rephrased as "*Analysis using ERA reanalysis data indicates that the July heatwave event was mainly located on the Pamir Plateau*"

53. L336: an equilibrium cannot be positive. Rephrase to: 'glaciers were previously considered in equilibrium or having a positive glacier mass balance', or similar.

Response: We clarify the expression following your suggestion.

54. L358: what are Rossby wave trains? Please add references to this statement.

Response: We added Hood et al., 2020 as the reference here.

Hood, L. L., Redman, M. A., Johnson, W. L., and Galarneau, T. J.: Stratospheric Influences on the MJO-Induced

Rossby Wave Train: Effects on Intraseasonal Climate, J. of Climate, 33, 365-627 389, 10.1175/JCLI-D-18-0811.1, 2020

55.  L360-362: this statement is more appropriate for the introduction.
Response: It was moved to the introduction.

**Grammar and typos:**

56.  L18: remove 'the' before heatwaves
Response:   According to the scope of this study, the sentence was revised to "This study analyzes the characteristics of daily glacier mass balance and their responses to the 2022 heatwaves based on time-lapse photography".

57.  L20: typo 2019/2000
Response: Sorry for the mistake. we changed 2019/2000 to 2019/2020

58.  L20: consider using 'and' instead of 'but'. When using 'but', you would expect mass gain for the other 2 years.
Response: We changed 'but' to 'and'.

59.  L23: heatwaves
Response: Done.

60.  L24: Glacier Loss Day, remove mass
Response: according to RC2's comment, the term "Glacier Loss Day" was removed.

61.  L25: wakened → weakened
Response : Done.

62.  L27: remove 'the' before heatwaves
Response: The original sentence was removed according to RC2's comment.

63.  L32: occurred → observed
Response: Done.

64.  L34: Hewiit → Hewitt
Response: Done.

65.  L35: maybe be  →  might
Response: Done.

66. L37: are 'a' critical component

Response: Done.

67. L41: 'the' Tibetan Plateau

Response: Done.

68. L55: Traditional glaciological method and geodetic survey → The glaciological method and geodetic mass balance method

Response: Done.

69. L56: dynamics → differences

Response: the sentence was revised to "*However, both approaches face challenges in capturing the high temporal evolution of surface mass balance.*"

70. L60: add 'and' before terrestrial laser scanning

Response: We added "and" before terrestrial laser scanning

71. L61: remove 'the' before extreme melt events

Response: 'the' was removed.

72. L63: developed → introduced

Response: this sentence was revised to "*New indices like the Glacier Loss Day (GLD), which was defined as the day when net mass balance becomes negative and winter snow is exhausted (Voordendag et al., 2023), have provided new insights into short-term mass balance variations.*"

73. L106, 107, 144, 148-151, 195: hight → height, but this actually needs to be the melt out length of the stake.

Response: Thanks for the suggestion, we used "stake melt-out length" in the revised version.

74. L118: S channel → S-channel

Response: "-" was added.

75. L145: dynamics → changes.

Response: the sentence was removed according to comment 19.

76. L158: glacial → glacier

Response: the part was removed.

77. L177: Fig 1 → fig 1b

Response: Done.

78. L207: "which own significant mass loss". Please clarify/rewrite.

Response:   According to comment 29, we replotted this figure using different color for each hydrologic year.

79. L239: the corresponding grid → add "point"

Response: Done

80. L252: add a reference to Fig. 8b after the radiation values.

Response:   we added "Fig. 7b (new Figure)" after the radiation values.

81. L317: analysis → analyses

Response: Done

82. L319: remove ground.

Response: we removed "ground".

83. L320: Ata regions → region.

Response: we changed "regions" to "region".

84. L328: remove the extra point after region.

Response: we deleted the extra point.

85. L330: replace dynamics with response.

Response: we changed "dynamics" to "changes".

86. L352-354: repetition of the previous sentence. Also, see comment 45.

Response: we deleted the repeated sentences.

87. L365: abbreviation TP is not introduced before. Tibetan Plateau, right?

Response: we changed 'TP' to 'the Tibetan Plateau'.

88. L376: "spring accumulation-summer ablation". Check spelling with hyphens here and make it the same throughout the paper. Is this type invented by the authors or have other glaciers similar behaviour?

Response: Fujita and Ageta (2000) used "winter-accumulation-type" and "summer-accumulation-type". Maussion et al., 2014 used "spring-accumulation type". The hyphen was placed between the seasonal term (e.g., spring/summer) and accumulation/ablation. We revised the sentence as: "generally characterizing glaciers in the eastern Pamir as *"spring-accumulation and summer-ablation"* type.". Previous research (Xu et al., 2017) shows that Urumqi Glacier No. 1 in the eastern Tien Shan has a similar behavior with the Kangxiwa Glacier.

Fujita K and Ageta Y.: Effect of summer accumulation on glacier mass balance on the Tibetan Plateau revealed by mass-balance model, J. Glaciol., 46(153), 244-252, 2000.

Maussion, F., Scherer, D., Mölg, T., Collier, E., Curio, J., and Finkelnburg, R.: Precipitation Seasonality and Variability over the Tibetan Plateau as Resolved by the High Asia Reanalysis, J. Climate, 27, 1910–1927, https://doi.org/10.1175/JCLI-D-13 00282.1, 2014.

Xu, C., Li, Z., Wang, F., Li, H., Wang, W., and Wang, L.: Using an ultra-long-range terrestrial laser scanner to monitor the net mass balance of Urumqi Glacier No. 1, eastern Tien Shan, China, at the monthly scale, J. Glaciol., 63(241), 792-802, 2017.

89. L379: brought forward? Please rephrase. For example, the GLD occurred one month earlier in 2022 compared to the other three years, due to exceptional melting early in the season and/or under average precipitation amounts in winter.

Response: we rephrased this sentence as "*Coupled with below-average winter–spring accumulation, these heatwaves pushed the equilibrium line altitude above the glacier's maximum elevation.*"

90. L383: replace deficit by loss.

Response: According to RC2's comment, the original sentence was revised to "*Our findings clarify their vulnerability to short-term climatic extremes and validate a practical surface mass balance monitoring method for remote mountain regions*"

---

## Author Comment (AC2)

The manuscript is reasonably well written and the figures are clear and generally informative, if not a bit redundant at times. The work is of sound scientific quality for the most part, but I struggle to find clearly-driven or stated research questions and support for a few statements in the manuscript. I believe that the relevance of the work needs to be more strongly supported in relation to the past research in the region. Moreover, a clearer presentation and discussion of the magnitude of summer heatwave events during the last decades is required, especially in the context of its role during the anomalous periods of stable or positive mass balance years. A sensitivity to an extreme heatwave year (2021-2022) alone is not suggestive of a termination of the Karakoram-Pamir anomaly as the authors state in their abstract. In several places, there are also unclear statements or those that are not clearly backed by robust tests (e.g. the relative role of precipitation and accumulation on the glacier compared to melt events). Ultimately, the relevance of the presented work needs to be placed better into the context of the broader mass balance patterns and changes over the last decades and considering the role of precipitation and snowfall/albedo. The authors should also work hard to improve the clarity of the text and spelling in several places. Based upon my comments below, I suggest major revisions before the manuscript could be accepted by the journal.

Response: Thank you for your time and constructive comments on the manuscript. We considered each comment carefully and incorporated practically all of them in the revised manuscript. In terms of research motivation and objective, we have added detailed descriptions of the research objectives into the last paragraph of the Introduction. And we also added the past research in the region, in particular the regional mass balance in the discussion Section. Following your suggestion, we did not mention the termination of the Karakoram-Pamir anomaly in the revised manuscript. We added an analysis of the impacts of precipitation variability on glacier ablation intensity to the Result and Discussion section, in particular the precipitation phase. Incorporating the community comment, we compared the mass balance variations of adjacent glaciers and added relevant discussion. We revised the spelling errors and unclear expression. Below are our point-by-point responses (blue color) and the changes we've made to the paper (in italic).

**Major comments**

**1. Research Questions**: The manuscript provides no clearly stated research questions that help to direct the main goal of the analysis and its context. The title indicates that heatwaves form the main focus of the paper and timelapse imagery is the tool by which it is evaluated, but the focus falls upon a single extreme year with no historical context regarding heatwave occurrence (or inference from reanalysis etc). While most of the material to frame this is available in the introduction, the goals (or even hypotheses) of the manuscript need to be clearly motivated and established at the end of the introduction.

Response: We sincerely appreciate this insightful suggestion. We have reclarified the research objectives of this study at the end part of the Introduction. We highlight the critical the knowledge

gap regarding glacier responses to extreme heatwaves and address the advantages of time-lapse camera-based daily mass balance observations for high-resolution investigations of glacier response. We also revised the three main goals of our study, which include: (1) characterizing the contrasting seasonal mass balance patterns of the Kangxiwa Glacier in the eastern Pamir across the 2019/2020 to 2022/2023 balance years under varying climatic forcing regimes; (2) quantifying the sensitivity of surface mass balance to the extreme 2022 summer heatwaves; and (3) identifying the atmospheric circulation anomalies linked to the 2022 heatwaves using ERA5 reanalysis data, and further elucidate regional glacier response heterogeneity. Please see the following description and more details in the introduction.

*"In the summer of 2022, heatwaves swept across many parts of the Northern Hemisphere, causing extreme heat events in North America, Europe, and the Yangtze River in China (Lu et al., 2022). However, the processes and mechanisms through which these heatwaves impact glaciers in the Eastern Pamir—long considered climatically stable—remain poorly constrained. To address this critical knowledge gap, this study prioritizes time-lapse camera observations to capture daily surface mass balance, complemented by in-situ stake measurements for cross-validation, and integrates ground-based meteorological station records and reanalysis datasets to interpret the associated climatic contexts and underlying mechanisms. Based on these high-resolution daily surface mass balance datasets, this study aims to (1) characterize the contrasting seasonal mass balance patterns of the Kangxiwa Glacier in the Eastern Pamir across the 2019/2020 to 2022/2023 balance years under varying climatic forcing regimes; (2) quantify the sensitivity of surface mass balance to the extreme 2022 summer heatwaves; and (3) identify the atmospheric circulation anomalies linked to the 2022 heatwaves using ERA5 reanalysis data, and further elucidate regional glacier response heterogeneity."*

**2. Relevance and context within an anomalous region:** The authors present some new understanding in a relatively under-studied region of anomalous glacier behaviour, but focus largely upon the role of temperature during summer heatwaves in one year toward affecting mass balance. Nevertheless, in such a region where winter and spring accumulation regimes dominate, the role of antecedent precipitation, accumulation and surface conditions will also have a strong impact on the annual mass balance. The variability of precipitation is one of the suggested mechanisms that was driving the mass balance anomaly for the region (e.g. Farinotti et al., 2020; de Kok et al., 2018) that has now likely ended (e.g. Hugonnet et al., 2021; Jouberton et al., 2025). However the authors present an unclear case for the role of precipitation variability in the winter and spring and, again, provide little broader context related to historical changes. Additional discussion and reframing of the manuscript are needed to provide a more valuable insight for the reader.

Response: We also appreciate this insightful suggestion. We acknowledge that precipitation constitutes another critical factor influencing regional glacier mass balance, particularly the cascading feedback effects of antecedent precipitation, which modulates the albedo feedback mechanism during the summer season. Quantitative assessment of this relationship via sophisticated modeling approaches lies beyond the scope of the present study. We have supplemented an analysis

of its impacts on summer glacier mass balance in the Results section, noting that the intense ablation observed in the summer of 2022 was also linked to the variations in the timing of heatwave occurrence and the combination with seasonal distribution of precipitation. The present study focuses specifically on the impacts of summer extreme high temperatures on 2022 glacier ablation, with a comparative analysis against the other three reference years to underscore this core research focus. Please see the details in Discussion 5.1.

**3.    Uncertainties:** The estimation of uncertainties and use of the additional mass balance stake observations (i.e. not at the timelapse camera locations - Figure 1) needs more clarification and justification in the study. The glacier-wide mass balance is computed from a weighted average of the three-point mass balance derived at the location of the timelapse cameras, based on their elevation and the hypsometry of the glacier. I do not understand if and how the 10 mass balance stakes are used to help constrain the extrapolation to the whole glacier, especially the 7 stakes which are not located near the cameras. I do not see anywhere that reports the mass balance derived from all of the stakes, which are presented in the map of the study site (Fig. 1d). The differences in mass balance retrieved at the three cameras, shown in Figure 6, highlight the large mass balance variability with elevation, which is not necessarily linear. Therefore, I am wondering how this simplified extrapolation affects the glacier-wide mass balance results and the results of the study in general. The three locations will have different sensitivity to the heatwaves, depending on the baseline temperature and its proximity to the freezing point. The relevance of avalanches and other re-distribution processes is not mentioned or discussed and the authors do not subsequently consider any of the uncertainties in their main results when citing key numbers.

Response: All three reviewers have highlighted this critical issue. In response, we have supplemented the relevant descriptions in the Methods section and added a new subsection (**4.1 Performance of glacier mass balance estimation based on time-lapse camera observations**) to the Results chapter of the revised manuscript. Glacier-wide mass balance was derived via a weighted average of the three pointing mass balance observations by cameras and their representative elevation-area. Although there are only three cameras available, their spatial representativeness was carefully considered before the installation near the terminus (5005m asl), middle (5137m asl) and upper region (5300 m asl) of Kangxiwa Glacier. The location of camera 1 at 5005m roughly represents the zone of 5005±60 m in elevations (~4960-5080m asl); Camera2 represents the zonal range of 5137±60 m in elevations (~5080 to ~5200 m asl); Camera3 represents the zonal range of 5300±100 m in elevations (~5200 to ~5390 m asl). Actually, the topography condition of Kangxiwa Glacier is relatively flat and the 10 stake mass balance measurement evidenced that there are linear mass balance gradients (See the following Figure). It allows us to use the weighing method to calculate the glacier-wide mass balance. Please see the details in the revised manuscript in Method.

*Glacier-wide daily mass balance was then derived by area-weighted sum of point-scale mass balance estimates from the three cameras. A 30-m resolution SRTM DEM was employed to quantify the area distribution with its elevations. The cameras were deployed near the terminus, middle and upper region of Kangxiwa Glacier, allowing the entire glacier to be partitioned into three distinct zones centered approximately on each camera's installation site. These three zones corresponded to the elevation ranges of 4960–5080 m (area weighting factor: 0.33), 5080–5200 m (area weighting factor: 0.38), and 5200–5390 m (area weighting factor: 0.29). Notably, each zone aligns with one camera deployment location and accounts for roughly one-third of Kangxiwa Glacier's total area. Kangxiwa Glacier features relatively flat topography (Fig. 1), and mass balance measurements from 10 ablation stakes further confirmed a linear elevation-dependent mass balance. These characteristics justified the extrapolation of glacier-wide mass balance from the limited set of point observations. Finally, the camera-derived mass balance estimates were validated against results obtained via traditional glaciological methods based on the 10 ablation stakes across the glacier.*

[Figure]

Figure R1 The elevation distribution of point mass balances measured by 10 stakes over

2019/2020-2022/2023 hydrologic years

We have also included a new Figure 4 (see below) to illustrate comparisons between the field stake method and camera-based mass balance observations at both point and glacier-wide scales. These comparisons demonstrate the robustness of daily glacier mass balance calculations derived from three camera-based observation systems, which cover the lower, middle, and upper reaches of the glacierized area.

[Figure]

**Figure 4** The comparison of pointed mass balance (a) and glacier-wide mass balance (b) measured by using the stake method and the Time-lapse photography on the Kangxiwa Glacier.

**4.1 Performance of glacier mass balance estimation based on time-lapse camera observations**

*Figure 3 shows a comparison of the cumulative point mass balance estimates derived from the time-lapse cameras and the glaciological method at the three camera sites (5005 m, 5137 m and 5300 m) on the Kangxiwa Glacier during the period from 2019/2020 to 2022/2023 balance years. Figure 4a further presents the seasonal comparative performance of point-scale mass balance (winter and summer). Quantitative comparisons reveal that the mean seasonal mass balance differences between the two datasets at the three monitoring sits are −64, −13, and 2 mm w.e, with corresponding standard deviations of 193, 137 and 104 mm w.e., respectively. Over the entire observation period, the mean discrepancies between the two datasets across three points yield an overall mean of 26 ± 149 mm w.e.*

*At the glacier-wide scale (Fig. 4b), the camera-based seasonal mass balance data exhibit a robust linear correlation with the stake-based counterparts (e.g., $R^2 = 0.98$), with almost all in-situ stake-based values falling well within the uncertainty bounds of the camera-based estimates. The mean discrepancy between the two datasets is 62 ± 54 mm w.e., indicating high consistency between the camera-based approach and the glaciological method. This robust agreement not only validates the reliability of time-lapse camera observations for quantifying glacier mass balance but also establishes a solid foundation for investigating temporal evolution characteristics of glacier-wide mass changes for the Kangxiwa Glacier.*

**4. The role of precipitation phase and surface albedo changes in glacier mass balance sensitivity is neglected:** Figure 6 shows the derived daily glacier-wide mass balance and the time series of precipitation recorded at the nearby automatic weather station. The agreement in timing between precipitation events and increases in glacier mass balance can be clearly seen from the figure. However, it is unfortunate that little attention is paid in the manuscript to precipitation type, especially considering that snowfall replenishment and albedo reset are suggested to play a key role

in glacier mass losses (e.g. lines 278-281). Summer snowfall will be a function of precipitation and air temperature, and I expect substantial differences along the altitudinal gradient of the glacier, which is not apparent, as most of the latter analyses in the manuscript focus on glacier-wide mass balances. I would suggest that the authors look at precipitation phase types, leveraging their meteorological observations and applying simple phase partitioning schemes such as dual-temperature thresholds or wet-bulb based parametrizations (Ding et al. 2014), to look at the amount of seasonal snowfall for each hydrological year and per elevation, and include this as part of the discussion on the inter-annual variability of glacier mass balance and sensitivity to heatwaves. The authors highlight the role of surface albedo on ablation rates, with the example of the 3rd period of heatwave in 2022 experiencing similar melt rates as earlier heatwaves despite lower air temperatures, likely due to darker surface conditions. With daily photos available for the three sites, it could be worthwhile examining the number of days per summer with bare-ice/darker albedo conditions, and linking this, even qualitatively, to ablation rates.

Response: Following your suggestion, we quantified the rainfall and snowfall contributions at the three camera sites using the method proposed by Ding et al. (2014), coupled with meteorological observations from AWS4900 and the AWS at the Muztagh Ata observation station (3650 m asl). The air temperature and pressure lapse rates were set at 0.6 °C per 100 m and 7.6 Pa m$^{-1}$, respectively; additionally, the annual precipitation elevation gradient was 11.2 mm per 100 m, and the relative humidity lapse rate was 1.3% per 100 m. Figure R2 below presents the seasonal snowfall and rainfall totals for each hydrological year across the three camera locations. Results indicate that precipitation in the glacierized area is dominated by snowfall. This explains the strong temporal consistency between precipitation events and increases in glacier mass balance in original Figure 6. Even during the exceptionally warm summer of 2022, snowfall accounted for 93.7% of total precipitation at the glacier terminus (5050 m asl), 95.7% at 5137 m asl, and 99.1% at 5300 m asl. The scarcity of rainfall in the glacierized area confirms that precipitation phase transitions likely exert only a minor influence on the surface energy–mass balance. We have added the relevant description in the Results 4.2.

As noted in Major Comment 2, in regions dominated by winter–spring accumulation regimes, antecedent precipitation, accumulation dynamics, and surface conditions also exert a substantial impact on annual mass balance. This contributes to the modulation of surface albedo and, in turn, ablation rates, as exemplified by the third heatwave event of 2022. This event exhibited comparable melt rates to earlier heatwave episodes despite lower ambient air temperatures, a pattern likely attributable to reduced surface albedo (darker surface conditions). We also further address such mechanisms in the Result Section when we explain the possible reason why the third heatwave contribute to the intensive melting. Please see the Result Section 3.3 in the revised manuscript.

[Figure]

Figure R2. Seasonal snowfall and rainfall totals for each hydrological year across the three camera locations.

Additionally, we compared the number of bare ice days and summer mass balance at the 5005 m and 5137 m sites over the 2019/2020 to 2022/2023 hydrological years (Figure R3). There exists a certain negative correlation between the number of bare ice days and summer (June-September) mass balance. However, due to the limited number of data points, this study cannot perform a reliable statistical quantitative analysis of the relationship between these two variables.

In the revised manuscript, we highlighted in the Results section that the significant increase in bare ice days during the summer of 2022 contributed to substantial mass loss, attributable to the lower albedo of bare ice compared to snow. We have added the following sentences: "*At the 5005 m site, the number of bare ice days from June to September reached 42 in 2021/2022, compared to 4, 23, and 13 days for the 2019/2020, 2020/2021, and 2022/2023 hydrologic years, respectively. This marked increase in bare ice days enhances glacier surface energy absorption, which partly explained the substantial mass loss in the summer of 2022.*"

[Figure]

Fig. R3 Relationship between the number of days with bare ice surface and summer balance at the 5005 m and 5137 m points

**5. Redundancy and lack of clarity in figures:** Figure 2 presents the methodology of the automatic extraction of height changes from time-lapse photography, but could be completed with a scheme or additional indications to help the reader visualize the different lengths introduced in the methods (L, Lp, Lpv, Lpa), making this section easier to follow. Figures 3 to 6 are somewhat redundant, showing the daily evolution of surface changes at the cameras' locations in various ways and units. I do not think Figure 4 adds much to the manuscript; it is mostly an overlay of Figure 3's content, and the inter-annual differences can already be seen clearly in Figure 6.

Response: Following the reviewers' recommendations, we have added relevant annotations to the original Figure 2, deleted Figure 4, and provided the revised figures below. Corresponding revisions have been made to the subsequent Methods section in line with the updates to Figure 2. In response to Reviewer Comment 1, the original Figure 6 has also been revised and the original Figure 7 was moved to the supplementary material. Additionally, we have included two new figures: one illustrating the performance of glacier mass balance estimation based on time-lapse camera observations, and another depicting the unprecedented summer warming event in 2022 in the eastern Pamir. Please refer to all updated figures in the revised manuscript.

[Figure]

Figure 2. Illustration of image processing. (a) The original image with a frame. The bottom part of the stake is shown in a magnified view. (b) the contour of the scale stake with the minimum bounding rectangle in red; (c) the grayscale representation of the S- channel.

**6. Clarity of the structure of the manuscript:** A substantial part of the manuscript introduces the set-up used to monitor glacier mass balances. The daily mass changes derived from time-lapse imagery are presented in the methods, but they could equally be shown in the results section. While both options would probably be fine, an outline of the paper's structure could be given at the end of the introduction, which is commonly done to guide the reader through the paper.

Response: We have revised the methodology section and relocated certain descriptive content to the results section. Additionally, we have explicitly addressed the knowledge gaps and core objectives of this study in the introduction. To comply with the guidelines of The Cryosphere, we have omitted the outline of the paper structure in the revised manuscript. Furthermore, both the title and key research objectives have been refined to center on the 2022 heatwave event, rather than long-term heatwave.

**Minor comments**

1.	Title: "In-situ Measurements" is quite broad, especially since Time-lapse photography can be considered as one of the in-situ measurements. Consider replacing it with something more specific (e.g., meteorological observations) or remove.

Response: We removed "In-situ Measurements" from the title. Regarding the key method of Time-lapse photography for high-resolution daily mass balance estimation, the revised title was changed as "*Glacier Mass Balance and Its Response to the 2022 Heatwaves for Kangxiwa Glacier in the Eastern Pamir: Insights from Time-Lapse Photography*".

2. Line 15: "anomalous less changes" is unclear and should be re-formulated. Anomalously less negative mass balances?

Response: We revised as "*anomalously less negative mass balances*".

3. Line 17: scarcity of high resolution observations of what? What resolution? Please clarify and state more specifically.

Response: we mean temporal resolution. It was revised to "scarcity of high temporal resolution of mass change observations".

4. Line 18: "the heatwaves", remove "the" if you do not specify which heatwaves are referred to here.

Response: It was specified as "the 2022 heatwaves".

5. Line 20: "2019-2000" is probably a typo.

Response: Sorry, we changed "2019/2000" to "2019/2020".

6. Line 19: there are no meteorological observations at the glacier itself. Rephrase.

Response: we added "nearby" to the sentence. The revised sentence is "*This study analyzes the characteristics of daily glacier mass balance and their responses to the 2022 heatwaves based on time-lapse photography, ablation stake/snow pit measurements and nearby meteorological record collected at the Kangxiwa Glacier in the eastern Pamir.*".

7. Line 21: "Observations evidence" -> "Observations show"

Response: Done

8. Line 24: The concept of the glacier loss day is not yet a widely-adopted term and should be avoided for the abstract and rather be clearly stated what it means.

Response: Following your suggestion, we deleted the glacier loss day in the Abstract.

9. Line 25: spelling ("wakened" -> "weakened").

Response: Done

10. Line 27: Are there any glaciers not sensitive to heatwaves?

Response: We deleted the sentence.

11. Line 27-28: A strong mass balance response to a single year with heatwaves cannot be used alone to suggest the end of the regional anomaly that is decadal or multi-decadal. Please remove or rephrase to be precise about what it can show/suggest.

Response: We revised it as "*Our finding revealed that short-term heatwaves can trigger substantial glacier mass loss in the Easter Pamir, once considered climate-resilient, suggesting that "Pamir–Karakorum" anomaly is being challenged by the rising frequency of extreme heat events.*"

12.  Line 46-47: Specify "in this region".

Response: We added "*in this region*" at the end of the sentence.

13.  Line 50: remove "s" from "for examples".

Response: Done

14.  Line 54: The Oliver et al. reference describes marine heatwaves and is not really appropriate (also in the discussion). The authors should try and provide some numbers for heatwaves in High Mountain Asia. Here they describe extreme events in general, but this can also be precipitation extremes, snow accumulation extremes or others that can affect glaciers. Please be specific regarding the characterisation of heatwaves in the region, or clarify that no such data/studies are available if that is the case.

Response: We have removed Olive et al. in the introduction and discussion. We added two references on heatwave impact on mountain glaciers in Apls and Tibetan Plateau. Following your suggestion, we only address the impact of heatwaves, rather than extreme events due to few such data and studies in the High Mountains.

Ref:

Colucci R., Giorgi F., Torma C., Unprecedented heat wave in December 2015 and potential for winter glacier ablation in the eastern Alps, Scientific Report, 7: 7090, DOI:10.1038/s41598-017-07415-1, 2017.

Zhu, F., Zhu, M., Yang, W., Wang, Z., Guo, Y., Yao, T., Drivers of the extreme early spring glacier melt of 2022 on the central Tibetan Plateau. Earth and Space Science, 11, e2023EA003297. https://doi.org/10.1029/2023EA003297, 2024.

15.  Line 63: Remove "d" from "Characterized".

Response: the sentence was revised as: "*Recent advancements in high-temporal-resolution monitoring techniques to monitor ablation, such as the SmartStake by rolling the steel wire up (A2PS contributors 2021), automated cameras monitoring colour-coded ablation stakes (Landmann et al., 2021; Cremona et al., 2023), and terrestrial laser scanning techniques (Voordendag et al., 2023), have provided new insights into short-term mass balance variations, including their response to extreme melt events (Cremona et al., 2023).*".

16.  Line 65-66: State clearly the research questions/hypotheses of the study here.

Response: We stated the research questions. *(1) characterize the contrasting seasonal mass balance patterns of the Kangxiwa Glacier in the eastern Pamir under varying climatic forcing regimes across the 2019/2020 to 2022/2023 balance years; (2) quantify the sensitivity of surface mass balance to the extreme 2022 summer heatwaves; and (3) identify the atmospheric circulation anomalies linked to the 2022 heatwaves using ERA5 reanalysis data, and further explore regional glacier response.*

17. Line 67: "Study regions" -> "Study region".
Response: Done.

18. Line 73: Any reference for this value of 70 mm of annual precipitation? From the AWS? It seems quite low (under-catch corrected?) and appears wholly inconsistent when compared to the accumulation values given later on in the manuscript.
Response: The mean value is from the Taxkorgan meteorological station (~3091 m a.s.l.) ~50 km south of Muztagh Ata), during 1960-2015. We also added the references in the revised manuscript.

Ref:
Li, Z., Wang, N., Chen, A., Zhang, L., and Zhang, Y.: Slight change of glaciers in the Pamir over the period 2000–2017, Arct. Antarct. Alp. Res., 54, 13–24, https://doi.org/10.1080/15230430.2022.2039766, 2022.
Lv, M., Quincey, D., Guo, H., King, O., Liu, G., Yan, S., Lu, X., and Ruan, Z.: Examining geodetic glacier mass balance in the eastern Pamir transition zone. J. Glaciol., 66, 260, 927-937. https://doi.org/ 10.1017/jog.2020.54, 2020.

19. Line 77: Are there no geodetic estimates of Hugonnet et al. (2021) for the specific glacier?
Response: We have added in the text "*Geodetic estimates indicate that the average mass balance of Kangxiwa Glacier was -0.13±0.99 m water equivalent (w.e.)*"

20. Line 84: You could specify the date and time at which the photo was taken
Response: We added the date of 20 June, 2020 in the caption.

21. Line 92: "900 kg/m$^3$", any justification or citation for this value? Why not follow the common values of the literature with uncertainty ranges?
Response: Following the suggestion and RC1'S comment, we added reference of Huss (2013) here.
Ref:
Huss, M.: Density assumptions for converting geodetic glacier volume change to mass change, The Cryosphere, 7, 877–887, 2013

22. Line 96-105: Can the authors provide information about the distance to the stake from the camera and the resultant pixel resolution?

Response: The distance between the cameras and the stakes was estimated to approximately 6-10 meters. We have added this information in the text. Unfortunately, the camera manual does not provide the information of focal length, so it is difficult to calculate the pixel resolution.

23.  Line 106: spelling mistake ("height, not "hight"). Here and in several places (Line 150 etc). Please revise and check carefully throughout the manuscript.

Response: Sorry for the spelling mistake. We have corrected them in the whole text.

24.  Line 143: "0.18 cm" reads like a very precise number. Make sure that you don't give numbers with a number of digits that goes beyond the precision achieved by your detection method.

Response: We have revised all the numbers and their precise. We did not mention this value in the revised manuscript. Centimetre-level precision was used for calculation.

25.  Line 152: How was the surface condition classified (snow vs ice)?

Response: The surface conditions were derived through visual inspection of the photos manually (snow/ice). This information was added in the revised text.

26.  Line 155: Do you have any justifications for the upper and lower bounds of snow density assumed? Could fresh snow fallen during very cold temperatures not have a density lower than 300 kg/m^3?

Response: The upper and lower bounds of snow density were adopted by in situ measurements of snowpack. The minimum and maximum snow density at the three sites during the four balance year were 286 kg/m$^3$ and 587 kg/m$^3$ respectively. We added the relevant information in the revised manuscript.

27.  Line 159: Can you be more explicit regarding what "lower" glacier mass changes mean here? Does it mean less negative, more negative, or less positive?

Response: It means less negative here.

28.  Line 160: "-409.02 mm w.e.", cf. comment above, I do not think the precision of your mass changes derived from the camera is higher than 1 mm. Consider changing to "-409 mm w.e."

Response: Thanks for pointing out this issue. We have rounded all the data in the revised manuscript.

29.  Line 165: I am not sure if a website link is acceptable, do you have a proper reference for the digital elevation model used?

Response: The SRTM DEM was used to calculate the area distribution of Kangwure Glacier. We did not provide the website in the revised manuscript.

30. Line 166: What about avalanches? Are they common on this glacier? Are these stakes really representative of those entire elevation bands? Following the major comment above, this needs to be critically considered when deriving uncertainties and at the very least, be discussed

Response: The topography condition of Kangxiwa Glacier is relatively flat (Figure 1) and the 10 stake mass balance measurement evidenced that there are linear mass balance gradients. There are no avalanches in the upper regions. We also compared both pointed and glacier-wide winter/summer mass balance calculated by using the traditional in-situ ablation method and the time-lapse photography on the Kangxiwa Glacier. Such comparison evidenced the reliability of the area-weight method by using three mass balance observations. We also added the relevant description to address the process in the Method section and their reliability of both point and glacier-wide mass balance in Result section in the revised manuscript (new Figure 4).

31. Line 167-168: Please clarify how the gaps were filled, as it is not clear from the description here in the manuscript.

Response: We have rephrased the description more clearly. The data gap was filled by using the interpolation method. And we have added this in the Section 3.2.3

*Limited data gaps were interpolated using adjacent measured values. The data gap at the 5005 m site from 1 October 2019 to 20 June 2020 was filled via daily mean mass balance value derived from the corresponding stake measurements.*

32. Line 180: Are these solid and liquid precipitation, or only liquid precipitation?

Response: It is all weather precipitation measured by T200B. We have added the information.

33. Line 196: How can the authors assert the role of sublimation from camera data alone at such a temporal resolution? Estimating sublimation requires specific measurements and modelling to characterize well.

Response: The low air temperature in winter prohibits the surface melting. The time lapse camera evidenced that the snow-cover surface was transferred to exposed ice surface condition, which means the possible surface sublimation or mechanical snow drift in the winter season. Therefore, the sentence was revised as "*The slight surface mass loss during this period was likely caused by sublimation or mechanical snow drift by strong winter wind, which was evidenced by the transition from snow-covered surfaces to exposed bare ice (Fig. S3)*".

34. Line 199: Explain what the ranges of value refer to. Are these the minimum and maximum annual values?

Response: We did not mention the detailed numbers any more in the revised manuscript. They mean maximum accumulation.

35. Line 209-210: Unclear. There are losses from the surface earlier in the season than the start of June (cf. Figure 6).

Response: Thank you for this comment. In the revised manuscript, we have clarified the onset of significant mass loss since the maximum net accumulation.

36. Line 213: specify the years corresponding to each value of the range "-400 to 957 mm" given.

Response: the sentence was revised as "*Another interannual variability lies in the magnitude of annual surface mass loss, with values of -414, -334, -907 and -694 mm w.e. in the 2019/2020, 2020/2021, 2021/2022 and 2022/2023 hydrologic years, respectively.*"

37. Line 219: "thermal conditions" -> "air temperature". To avoid confusion with the ice temperature.

Response: we used air temperature in the revised version.

38. Line 231: How does this interpolation compare to the additional stake data that do not have cameras (see major comment)? Is this the same evaluation that is described earlier in the manuscript (lines 140-145)?

Response: This part was revised as: "*Meanwhile, linear altitudinal interpolation of two complementary datasets- camera-based point mass balances from three elevations and stake-based measurement from 10 ablation stakes- enabled for the quantification of the equilibrium line altitude (ELA) over the study period. The ELAs displayed significant interannual fluctuations: the highest ELA surpassed the glacier summit during 2021/2022 balance year, whereas the lowest ELA (5079~5086 m a.s.l.) occurred in 2019/2020.*"

39. Line 240: The highest what? Be specific.

Response: It means the highest air temperature. The revised sentence was "*The ERA5 temperature data confirmed that the corresponding grid point of the Kangxiwa Glacier in July 2022 was the highest recorded between 1981 and 2024*".

40. Line 267-268: "Mass loss intensity […] was even greater than those during the two July heatwaves" -> This does not seem to be consistent with the numbers given in the last column of Table 1, please check this.

Response: The Table 1 present daily anomalies of mass balance during the three heatwave periods in 2022 relative to the corresponding 2020–2021–2023 average. The value "23.3" is the average daily mass balance for the period of August 5–17, 2022. According to RC1'suggestion, the table was removed.

41. Line 269: Spelling error for "Given".

Response: We corrected the spelling error.

42.  Line 269: What is meant by "reduced atmospheric radiations?" Shortwave? Clarify and write clearly and precisely.

Response: It means shortwave. The sentence was revised to "*Given the limited magnitude of warming and the reduced shortwave radiations*"

43.  Line 269-271: Figure S2 actually shows fewer ice exposures during this year compared to 2022/2023. How can the authors reconcile this with their suggestion of albedo being the main driver (cf. line 279-281)?

Response: We apologize for the graphical errors. Specifically, we failed to use the latest data for graphing, and the inappropriate plotting method employed in Python, resulted in incorrect visualization of ice/snow cover. According to the revised graph, the duration of bare ice in the summer of 2023 was shorter than that in 2022.

[Figure]

Figure S3 Changes in stake melt-out length derived by subtracting all data from the baseline values recorded on the first day of each hydrologic year on the Kangxiwa Glacier and evolutions of snow and bare ice surface conditions over the hydrological years 2019/2020 to 2022/2023.

44.  Line 277: A range of 100 mm is substantial for the region, but again this number and the range is notably larger than the value reported earlier in the manuscript. Please clarify those differences.

Response: Precipitation amount (70 mm) in the "Study region" refers to data from low-elevation Taxkorgan (at an elevation of 3091 m a.s.l.) and serves to demonstrate the dry climate background

of the eastern Pamir. The precipitation data utilized to analyze climate condition of the Kangxiwa glacier was monitored by the AWS at 4900 m a.s.l., with precipitation for the 2019/2020, 2020/2021 and 2022/2023 hydrological years around 300 mm. This discrepancy could be attributed to the increase in precipitation with elevations.

45. Line 278: How can the precipitation amounts not explain the differences? How specifically was this tested? Do the authors simply refer to the relationship in Figure 10? What about antecedent conditions and precipitation in the month prior? Once more, the focus falls on heatwaves, but one year and ~3 months of the analysis see heatwaves, whereas the discussion of the paper talks more broadly about the anomalous glacier region and how this might support the end of the anomaly. Ultimately, more care is required in assessing the related role of precipitation and snowfall.

Response:    As reply to Major Comments, we revised this paragraph as follows.

*Glacier surface melting is critically linked to the energy supply and the surface conditions. The precipitation phrase may greatly influence the snow accumulation and the surface albedo condition (Jouberton et al., 2022). Seasonal snowfall and rainfall amount at each camera-monitored site for each hydrological year were estimated using the method proposed by Ding et al. (2014), combined with meteorological data from AWS4900 and the Muztagh Ata observation station (3650 m a.s.l.). Snowfall accounted for 94% of total precipitation at the glacier terminus (5050 m a.s.l.), 96% at the mid-glacier site (5137 m a.s.l.), and 99% at the upper-glacier site (5300 m a.s.l.) in the warmest 2022. The scarcity of rainfall in summer season across the glacierized area indicates that precipitation phase transitions likely played a limited role in modulating the surface energy–mass balance. At the 5005 m site, the number of bare ice days from June to September reached 42 in 2021/2022, compared to 4, 23, and 13 days for the 2019/2020, 2020/2021, and 2022/2023 hydrologic years, respectively. This marked increase in bare ice days enhances glacier surface energy absorption, which partly explained the substantial mass loss in the summer of 2022. Given the moderate temperature anomaly and reduced shortwave radiation during the August heatwave, the intensive mass loss could be primarily attributed to the exceptionally low surface albedo of exposed ice (Fig. 8), which amplified solar radiation absorption and subsequent melt processes (Mölg et al., 2014).*

46. Line 304-305. This is a key statement, but make sure you keep here the link between accumulation variability during the accumulation period and the mass losses in the ablation season. The surface albedo changes will be a function of the seasonal snowpack height, such that the ablation rates are also linked to differences in accumulation (again related to the previous point and the major comment). Also, how have these contribution percentages been calculated?

Response: The original contribution percentages were calculated based on the mass change values and were removed. The relevant statement was incorporated into the Discussion section as following:

*Furthermore, variations in the timing of heatwave occurrence and their combination with seasonal distribution of precipitation can also contribute to substantial disparities in the response of glacier surface mass balance to climatic conditions. As illustrated in Figures 5, the mass loss in 2019/2020 was concentrated in August (-346 mm), following a period of high accumulation (+445 mm). In contrast, the 2022 mass balance process featured low spring accumulation and extremely*

*strong summer ablation driven by heatwaves in July and August. While the ablation periods in 2021 and 2023 were significantly longer than in other years, the average summer ablation intensity was moderate. Unlike the stable mass accumulation observed in June across the other three years, the Kangxiwa Glacier experienced early ablation (-107 mm w.e.) in June 2023, followed by the highest ablation in September over the four-year period. Cumulative mass loss from June onward reached -628 mm w.e. in 2023, which was slightly less than the maximum loss in 2022. These divergent patterns of mass balance evolution not only underscore the complexity of glacier responses to climate change in this region but also highlight the critical importance of continuous, high-temporal-resolution monitoring of glacier surface mass changes to inform future model-based explanations (Barandun and Pohl, 2023).*

47. Line 290: How do glaciers initiate snow accumulation? Please reword.

Response: We rephase as "Barandum et al. (2018) found that glaciers in the West Pamir region received snow accumulation at the start of the hydrologic year and accumulated over 1 m w.e. of snow accumulation during the winter season."

48. Lines 306-313. This reads like a result sentence; consider moving to the result section.

Response: This part was used to discuss the complexity of precipitation/glacier mass balance seasonality and their climate response characteristics and was kept here.

49. Lien 317: Spelling - "stereo" - "analyse"

Response: We corrected the typos.

50. Line 319: 2022 was a positive year from the results of Falaschi? This is then in disagreement with the strong negative mass balances reported by this study therefore? Please clarify how the studies compare and elaborate on the reasoning for any strong contrasts.

Response: Sorry for the mistake. The number was incorrect. It is a negative number -0.17±0.22.

51. Line 342: The high glacier mass loss during the 2022 heatwave was also reported further west in the Pamirs (Jouberton et al. 2025).

Response: Thanks for the suggestion. We added one sentence in the manuscript as follow: "*Latest research (Jouberton et al., 2025) reported that enhanced glacier mass loss linked to the 2022 heatwave in the Northwestern Pamirs*".

52. Line 364: Anticyclone "development"

Response: we changed "developed" to "development".

53. Line 376: What are "regional glaciers"? Please rewrite.

Response: we revised "regional glaciers" to "eastern Pamir glaciers".

54. Line 377: Again what is the role of accumulation regimes within this context of heatwaves and how does this compare with other studies on the sensitivity of glaciers in the region to temperature vs. precipitation? The summer of 2022 also followed a period of anomalously low snow accumulation (April 2022), similar to the European winter of 2021/2022. How much influence did that have?

Response: We rectified the expression to "*Notably, short-term heatwaves could greatly enhance the mass loss and then dominate annual surface mass balance in this region. Three heatwaves in July–August 2022 induced over 800 mm w.e. of mass loss within 40 days. Coupled with below-average winter–spring accumulation, these heatwaves pushed the equilibrium line altitude above the glacier's maximum elevation*."

55. Line 378: What is extreme sensitivity? Relative to what? How/when are glaciers not sensitive to heatwaves? The authors need to be more precise and more scientific in their writing.

Response: we removed the sentence.

56. Line 381-383: This is clearer than what is written in the abstract, but still requires more substantiation in the manuscript. Where do we see evidence of increasing frequency of extreme weather in the East Pamir in the manuscript? Extreme in what sense? Only in heatwaves? Where is the evidence of this and are there any longer term mass balance data that can aid this interpretation of temperature changes and extremes driving this "transition"?

Response: We don't mention transition or the end of the "Pamir–Karakorum" anomaly in the revised version. We highlighted that glaciers in this region may undergo substantial mass loss in response to future extreme high-temperature events.

**Figures**

Figure 1: Consider replacing the studied area symbol by a more visible marker like a star or a triangle. It is a bit difficult to see in Figure 1a.

Response: A triangle was used as the studied area symbol. Following is the revised figure.

[Figure]

Figure 3: The legend could be misleading, as "manual" could be mistaken as a measurement done by a person physically on-site. Consider expanding the legend as "camera: automatic", "camera: manual", or similar.

Response: The legend was revised following the suggestion.

[Figure]

Figure 4: missing a legend for the shaded areas in the figure background.

Response: In the original figure, shaded aeras indicate three phases of glacier mass change: slightly mass change (October to mid-April), mass accumulation (mid-April to June), and ablation (June to September). According to Major comment 4 and RC1's comment, we deleted the figure.

Figure 5: Consider using different scales for the three sub-panels to increase the readability of daily accumulations in the upper two panels. Alternatively, have you perhaps tried to overlay the three curves with different colors in a single panel?

Response: The three curves were replotted in one panel with different colors. Additionally, the filling data of the data gaps were shown with dotted line.

[Figure]

**Figure 3**. Comparison of the accumulated mass balance estimated using time-lapse cameras (lines) and the glaciological methods (triangles) at the three locations (5005 m, 5137 m and 5300 m) on the Kangxiwa Glacier. Dotted lines denote gap-filled data.

Figure 6: Write in the legend the meaning of the shaded areas around the mass balance lines.

Response: This figure has been redrawn according to RC1's suggestion. We used shaded area to present the mass change uncertain in the new figure below. The meaning of the shaded areas was included in the caption.

[Figure]

**Figure 5.** Cumulative glacier-wide mass balance of the Kangxiwa Glacier during the 2019/2020–2022/2023 hydrological years with the uncertainties by shaded area (a). The cumulated precipitation (b) and the cumulated positive degree day recorded by the AWS4900 (c).

Figure 8: In the caption, precisely state that the data shown in panels a-c comes from the AWS.

Response: We added one sentence "The meteorological data was derived from the AWS4900" in the caption (new Figure 7) "AWS4900" was abbreviation for Automatic weather station at 4900 m a.s.l. (section 3.3).

Figure 10: Any reason why the y-axis corresponding to precipitation is reversed? It might be more intuitive to have it the other way around.

Response: This figure was removed according RC1's suggestion.

Figure 11. This is a nice addition to your locally derived results, but it is unclear what the anomaly is relative to. Please clarify. I would recommend swapping the colorbar of Panel d, such that the red

colors correspond to higher melt conditions, as in the other three panels. Legend: Please specify the sources of the black outlines.

Response: The anomalies of meteorological variables and atmospheric circulations in July 2022 relative to the 1991-2020 mean and it was added to the caption. The color bar of Panel d has been swapped. Black lines represent the domain of the Tibetan Plateau

[Figure]

**Figure 9.** The anomalies of meteorological variables and atmospheric circulations in July 2022 compared with climatological (1991-2020) average. (a) Air temperature; (b) Geopotential height (unit: gpm) and wind field anomalies at 200 hpa; (c) Shortwave incoming radiation (SWin, unit: W/m2) anomaly; (d) Cloud cover fraction anomaly. Black lines represent the domain of the Tibetan Plateau (Zhang et al., 2021).

Ref:

Zhang, Y., Li, B., Liu, L., and Zheng, D.: Redetermine the region and boundaries of Tibetan Plateau, Geogr. Res., 40(6), 1543-1553, doi:10.11821/dlyj020210138 cstr:32071.14.dlyj020210138, 2021 (in Chinese with English abstract).

---

## Author Comment (AC3)

**Response to community comments by Dr. Martin Hoelzle**

This manuscript presents glacier mass-balance measurements from a glacier in the eastern Pamir mountain range. This region has recently been described as one of the few areas where glaciers have experienced relatively little mass loss and were thought to remain close to equilibrium conditions. However, the study suggests that the so-called Pamir–Karakoram Anomaly may be coming to an end, a conclusion the authors support with their mass-balance observations.The paper documents daily glacier mass-balance measurements from 2019 to 2023. It analyzes the characteristics of these daily mass-balance variations and their responses to heatwaves, drawing on time-lapse photography, ablation-stake and snow-pit measurements, as well as meteorological data collected at Kangxiwa Glacier in the eastern Pamir. The manuscript describes in particular the influence of heatwaves—most notably in 2022—on the glacier's behavior near the Muztagh Ata range. Overall, the paper is well structured and carefully prepared. The authors present well-developed mass-balance measurements, and the discussion offers a balanced and thoughtful analysis of the study's findings. I strongly support this paper for publication.

Response: Thank Dr. Martin Hoelzle for your time in reviewing our manuscript. We are grateful for all your comments. Below are our point-by-point responses (blue color) and the changes we've made to the paper (in italic).

There are, however, a few points that could strengthen the manuscript. In the Methods section, the authors should more clearly explain how the point measurements were interpolated to estimate mass balance across the entire glacier surface, and they should specify which interpolation method was used. These results should also be illustrated in a figure.

Response: Thanks for pointing out such issue which was also addressed by other reviewers. Following all reviewer's comments, we added more detailed description of how the three camera-based point mass balance data was interpolated to derive the glacier-wide mass balance. Glacier-wide mass balance was derived via a weighted average of the three pointing mass balance observations by Cameras and their representative elevation-area. Although there are only three cameras available, their spatial representativeness was carefully considered before the installation near the terminus (5005m asl), middle (5137m asl) and upper region (5300 m asl) of Kangxiwa Glacier. The location of camera 1 at 5005m roughly represents the zone of 5005±60 m in elevations (~4960-5080m asl); Camera2 represents the zonal range of 5137±60 m in elevations (~5080 to ~5200 m asl); Camera3 represents the zonal range of 5300±100 m in elevations (~5200 to ~5390 m asl). Actually, the topography condition of Kangxiwa Glacier is relatively flat and the 10 stake mass balance measurement evidenced that there are linear mass balance gradients (See the following Figure). It allows us to use the weighing method to calculate the glacier-wide mass balance. Please see the details in the revised manuscript in Method.

We also added one new Figure 4 (please see in following) to show the comparison between the field stake method and camera-based mass balance observations in both point scale and glacier-wide scale. Such comparisons evidenced the robustness of daily glacier mass balance calculation by

using three camera-based mass balance observations, which covered the lower, middle and upper glacierized region.

[Figure]

**Figure 4** The comparison of pointed mass balance (a) and glacier-wide mass balance (b) measured by using the stake method and the Time-lapse photography on the Kangxiwa Glacier.

In the Introduction and Discussion sections, it would be beneficial to compare the Kangxiwa Glacier measurements with other mass-balance records from nearby glaciers. Relevant examples include the Zulmart Glacier, Glacier No. 457 in the Pamir, as well as the Abramov Glacier. Additionally, the question arises as to why the newly measured data have not been submitted to the World Glacier Monitoring Service.

Response: Following your suggestion, we have compared the published annual mass balance from nearby glaciers including Zulmart Glacier, Glacier No. 457 in the Pamir, and Abramov Glacier. We found the Zulmart Glacier in the Pamir, which is about 200 Km northwestern from Kangxiwa Glacier, also exhibited the most negative mass balance in 2022. However, neither the Abramov Glacier nor the No. 457 Glacier showed such a pattern and the most negative mass loss occurs in 2021(Abramov Glacier) and 2023 (No. 457 Glacier). It evidenced that heterogeneous glacier mass changes in the Pamir and Tien Shan region. Our four-year observations also pointed out the fact that both the variations in the timing of heatwave occurrence and the seasonal distribution of precipitation can also contribute to substantial disparities in the response of glacier surface mass balance to climatic conditions. Although this study did not provide quantitative analysis by using sophisticated models, we highlight the complexity of glacier responses to climate change in Pamir and the critical importance of continuous, high-temporal-resolution monitoring of glacier surface mass changes to support future model-based explanations. We added the relevant discussion in Section 5.1.

Regarding the in-situ mass balance data, we will submit to the WGMS after the publication of this paper.

**Minor corrections:**

Line 25: change 'wakened' by 'weakened' or maybe better 'decreased'

Response: Done

Figure 1, Legend in the figure: change 'coutour' to 'contour'

Response: Done

[Figure]

Line 106 and everywhere in the paper: stake 'hight' should be replaced by stake 'height' or stake 'length'

Response: The stake melt-out length was used in the revised paper.

Line 186: Using ERA5 data, it should be also mentioned that these type of data still has remarkable uncertainties (e.g. Barandun and Pohl 2023).

Response: Thanks for pointing out this issue. The ERA5 data do have remarkable uncertainties, in particular for high mountain regions. Actually, the ERA5 data including geopotential height, wind fields, and surface air temperature, was mainly used to investigate the large-scale atmospheric circulation pattern. The regional air temperature change was mainly investigated by using the ground-based automatic weather station (AWS). Thus, we did not address the uncertainty of ERA5 in the manuscript.

Line 205, Figure 6: the lines shown in Figure does not correspond to the lines in Figure a, b and d. Please correct the lines that they have the same pattern.

Response: This figure has been replotted following the comments provided by you and the other reviewers. We have also included cumulative precipitation data for comparative analysis, and standardized the color scheme for different years across all sub-figures to enhance visual clarity.

[Figure]

**Figure 5**. Cumulative glacier-wide mass balance of the Kangxiwa Glacier during the 2019/2020–2022/2023 hydrological years, with the uncertainties by shaded area (a). The accumulated precipitation (b) and positive degree recorded by the AWS4900 (c).

Line 271, Table 1: It would be nice to show all radiation components individual meaning that we have columns for shortwave incoming, shortwave outgoing, longwave incoming and longwave outgoing radiation.

Response: Regarding the AWS is located in the non-glacierized region, the outgoing shortwave and longwave radiation did not therefore provide. Following the other reviewer's comment, the Table was removed in the revised manuscript.

Line 350-352: Sentence is duplicated. Please remove.

Response: Thanks. We have done it.